# Tauopathies: Deciphering Disease Mechanisms to Develop Effective Therapies

**DOI:** 10.3390/ijms21238948

**Published:** 2020-11-25

**Authors:** M. Catarina Silva, Stephen J. Haggarty

**Affiliations:** Chemical Neurobiology Laboratory, Center for Genomic Medicine, Departments of Neurology & Psychiatry, Massachusetts General Hospital and Harvard Medical School, Boston, MA 02114, USA; shaggarty@mgh.harvard.edu

**Keywords:** tau, aggregation, neurodegeneration, Alzheimer’s disease, frontotemporal dementia, pathogenicity, therapeutics, immunotherapy, tau degrader

## Abstract

Tauopathies are neurodegenerative diseases characterized by the pathological accumulation of microtubule-associated protein tau (MAPT) in the form of neurofibrillary tangles and paired helical filaments in neurons and glia, leading to brain cell death. These diseases include frontotemporal dementia (FTD) and Alzheimer’s disease (AD) and can be sporadic or inherited when caused by mutations in the *MAPT* gene. Despite an incredibly high socio-economic burden worldwide, there are still no effective disease-modifying therapies, and few tau-focused experimental drugs have reached clinical trials. One major hindrance for therapeutic development is the knowledge gap in molecular mechanisms of tau-mediated neuronal toxicity and death. For the promise of precision medicine for brain disorders to be fulfilled, it is necessary to integrate known genetic causes of disease, i.e., *MAPT* mutations, with an understanding of the dysregulated molecular pathways that constitute potential therapeutic targets. Here, the growing understanding of known and proposed mechanisms of disease etiology will be reviewed, together with promising experimental tau-directed therapeutics, such as recently developed tau degraders. Current challenges faced by the fields of tau research and drug discovery will also be addressed.

## 1. MAPT and Tauopathy Spectrum Disorders

Over a century after its first described case, Alzheimer’s disease (AD) is the most prevalent form of tauopathy and the most common cause of dementia (~60–80% of cases), and its frequency of incidence is rapidly increasing as the world’s population aged >65 continues to increase. Approximately 5.8 million Americans lived with AD in 2019, and this is predicted to double by 2050 [1,2], together with a financial burden predicted to increase from its current annual US $259 billion to more than $1 trillion by 2050. This trend is predicted to be global unless means of delaying, preventing, or treating AD are found [1,3].

The microtubule-associated protein tau (MAPT) is a neuronal protein that regulates microtubule stability and dynamics as well as axonal transport [4,5]. Tau binds to microtubules via repeat microtubule-binding domains in the C-terminus, and this process is regulated by phosphorylation of sites within and adjacent the binding region (Figure 1a,b) [6]. The N-terminal projection region plays a role in signal transduction and membrane interactions (Figure 1a) [6]. Other tau physiological functions include interaction with the plasma membrane and scaffold proteins, signal transduction, DNA/RNA protection, and regulation of synaptic function [7,8]. In the human central nervous system (CNS), six tau isoforms are expressed by alternative splicing of the *MAPT* exons 2, 3, and 10, of which the longest isoform 2N4R tau (441 amino acids) contains two N-terminal inserts and four repeat domains in the C-terminus region (Figure 1a) [9]. This process is developmentally regulated and specific to each brain region based on physiological function [10,11]. Exons 2 and 3 are translated into the N1 and N2 domains, respectively, producing the 0N, 1N, and 2N tau isoforms of the N-terminal projection region (Figure 1a). In the human adult brain, the 2N isoform is the least expressed while the 1N isoform is the most abundant [10]. Exon 10 encodes the second microtubule-binding repeat domain in the C-terminal region (Figure 1a). Inclusion of exon 10 leads to the expression of three tau isoforms with four microtubule-binding domains (4R-Tau), whereas exclusion of exon 10 leads to expression of three isoforms of 3R-Tau [10,12]. These four repeat domains (R1–R4, Figure 1a) are essential for tau ability to regulate stability of microtubules and support axonal transport. For this reason, relative 3R/4R expression is also developmentally regulated. During the fetal stage, 3R-Tau (0N3R) is the main isoform present, allowing for dynamic axonal properties conducive to synaptogenesis and formation of neural pathways, followed by postnatal expression of all isoforms. In the adult brain, 4R-Tau binds more tightly to microtubules and the overall 3R/4R ratio is maintained at 1:1 [10,11]. Despite its protein domains, tau’s native state defies the traditional ‘structure-function paradigm’ by lacking a well-defined three-dimensional structure, being classified as an intrinsically disordered protein. This is a characteristic of proteins that require rapid conformational changes and structural plasticity but is also a characteristic of proteins with high propensity for misfolding that play a role in the pathogenesis of neurodegenerative diseases [13,14]. Tau misfolding and aggregation into highly ordered β-sheet-rich paired helical filaments (PHFs) that subsequently deposit in the form of neurofibrillary tangles (NFTs) (Figure 1b) are implicated in a heterogeneous group of aging-related neurodegenerative disorders referred to as tauopathies, which include Alzheimer’s disease (AD), Pick’s disease (PiD), frontotemporal dementia (FTD), and progressive supranuclear palsy (PSP) (Table 1) [15,16,17,18,19,20,21,22,23,24,25,26,27,28,29]. While many *MAPT* mutations increase tau’s propensity for aggregation and toxicity, and are the cause of dominantly inherited tauopathies [30], the majority of tauopathies are sporadic with variable clinical and pathological presentations [15]. Tauopathies are mainly considered gain-of-function proteinopathies but, despite increasing understanding of tau physiology and role in disease, the mechanisms of tau aggregation with disruption of molecular pathways leading to neuronal death are still poorly understood [31,32,33]. Evidence indicates that native tau is highly soluble, contains several charged and hydrophilic residues, and shows little tendency for aggregation. Thus, for tau to become aggregation competent, it must undergo conformational and post-translational modifications (PTMs) within and near the hexapeptide motifs in the C-terminal repeat domain (Figure 1b,c) [34,35], which also makes 4R-Tau more aggregation prone [36,37]. Little is known about the consequences of tau loss-of-function, but reduced binding of hyperphosphorylated tau to axonal microtubules may alter their structure and/or function, disrupting axonal transport, driving synaptic dysfunction and loss, and promoting neurotoxicity.

Diseases where tau has a direct and predominant causal effect on neurodegeneration are referred to as ‘primary tauopathies’, which include progressive supranuclear palsy (PSP), corticobasal degeneration (CBD), Pick’s disease (PiD), aging-related Tau astrogliopathy (ARTAG), argyrophilic grain disease (AGD), primary age-related tauopathy (PART), and tangle-only dementia (TOD) (Table 1) [38,39]. Other amyloidosis that are also associated with the formation of tau inclusions, but where tau is not the primary or unique pathological feature, such as AD and chronic traumatic encephalopathy (CTE), are referred to as ‘secondary tauopathies’ (Table 1) [39,40]. The distinction between the two categories does not imply that there is not an equally important role for tau in the pathophysiology and relevance for tau-directed therapeutics. Although beyond the scope of this review, Table 1 summarizes key aspects that highlight the complexity and diversity of disorders associated with tau pathology. To date, there is no cure or disease-effective treatment that targets the cause of any tauopathy [41].

Approved symptomatic therapeutics include acetylcholinesterase inhibitors and memantine for AD to treat cognitive and behavioral symptoms, levodopa or dopamine agonists for FTD-associated parkinsonism motor dysfunction, and antidepressants (e.g., selective serotonin reuptake inhibitors). Current research has shown progress on different strategies to mitigate tau accumulation, prevent aggregation, and promote clearance [41,42]. Growing evidence suggests that early Tau PTMs, misfolding and oligomerization, impaired protein degradation, and Tau relocalization have higher impact on toxicity than late-stage PHFs and NFTs. Based on this, multiple experimental therapeutic approaches focus on targeting early forms of toxic tau and in promoting enhancement of protein clearance. So far, the strategy showing the greatest progress as measured by advancement into clinical trials is tau immunotherapy, where humanized tau antibodies have reached clinical trials for AD, PSP, and PPA. Even so, a major roadblock for therapeutic development is the still incomplete understanding of the molecular mechanisms and pathways involved in tau-mediated neuronal toxicity and death, which constitute probable therapeutic targets. In an effort to connect the knowledge from these two research fields, here we will review the current and ever-evolving understanding of the mechanisms of tau pathogenicity and respective approaches to therapeutics development.

## 2. Molecular Mechanisms of Tau Pathology

There are two non-mutually exclusive principal models for the mechanism of tau-induced neuronal pathology [43,44]. One model focuses on tau’s propensity for misfolding, oligomerization and fibril formation, and toxic gain-of-function, which is exacerbated by tau PTMs and inefficient clearance (Figure 2) [45,46]. The other model focuses on tau loss-of-function as a result of abnormal PTMs and sequestration into aggregates, leading to disruption of axonal integrity and transport (Figure 2) [43,47]. Here, we will review several molecular mechanisms and protein modifications associated with tau-mediated toxicity in the CNS, which represent potential therapeutic targets of tauopathy.

### 2.1. Protein Post-Translational Modifications

Tau PTMs, such as phosphorylation, acetylation, ubiquitination, and SUMOylation regulate Tau function and degradation, via temporal and regional regulation of the protein affinity for microtubules [35,48]. Tau has more than 80 phosphorylation sites reported [47,49], 23 acetylation sites [50,51,52], and several putative ubiquitination sites [45]. However, PTMs are also associated with tau pathology (Figure 1c). Increases in tau negative charge by phosphorylation or removal of the lysine positive charge by acetylation has a significant effect on tau dissociation from microtubules [53] and increases its propensity for self-oligomerization and fibril formation, as evidenced by hyperphosphorylated tau being the primary component of NFTs. Additionally, high MW soluble P-tau species, but not fibrillary tau, have been referred to as “bioactive forms” that can be released by neurons and taken up by neighboring cells, contributing to abnormal tau spreading and templated misfolding [54].

Phosphorylation of tau at several Ser/Thr residues, including Ser262, Ser356, and Ser396, is detected at basal levels and is likely relevant for the regulation of tau localization and binding to tubulin [55,56]. However, several other tau phospho-sites are absent during brain development or in cognitively healthy adults but are found enriched in disease [57,58]. Phosphorylation of Ser422 is almost never detected in cognitively healthy adults or in the fetal brain [59] but is detected early in AD [60] and shows high correlation with loss of cholinergic neurons and cognitive impairment [61]. Therefore, protein kinases involved in the pathological phosphorylation of tau at Ser422 could represent promising therapeutic targets. One example is the CNS-specific tau tubulin kinase TTBK1 that phosphorylates tau at several phospho-epitopes, including Ser422 (Figure 1c) [62,63,64]. Perhaps one of the best-characterized phospho-sites for tau is the Ser202/Thr205 (recognized by the AT8 antibody, Figure 1c), which drives tau fibril formation and is used to assign Braak stage based on a high occurrence in NFTs and high correlation with postmortem pathology [65]. In addition, phosphorylation at the C-terminus Ser396/Ser404 (Figure 1c) is one of the earliest events in AD preceding the formation of tau fibrils [66]. Antibodies towards this epitope (e.g., PHF1) have been widely used to follow tau aggregation and NFT formation. Additional phosphorylation sites include Ser208 in AD, which is found at 3x the normal levels in CSF, affecting tau function and driving aggregation [67,68]. This phospho-epitope is also found in the brain of PS19 (Tau-P301S) and rTg4510 (Tau-P301L) transgenic mouse models, in PSP tufted astrocytes, and in CBD astrocytic plaques [67]. A recent study by Dujardin et al. [69] revealed that the rate of AD progression is correlated with specific tau phospho-epitopes, namely Thr231/Ser235 and Ser262 (Figure 1c), which also correlate with tau seeding activity. Several kinases and phosphatases have been shown to regulate tau phosphorylation in physiological and disease contexts, and an imbalance in kinase/phosphatase activity is believed to contribute to the accumulation of P-Tau in disease. This has been extensively reviewed by others [70,71]. Briefly, tau kinases include proline-directed glycogen synthase kinase-3 (GSK-3), cyclin-dependent kinase 5 (CDK5), and 5′ adenosine monophosphate-activated protein kinase (AMPK); non-proline-directed kinases casein kinase 1 (CK1), microtubule affinity-regulating kinases (MARKs), and cyclic AMP-dependent protein kinase A (PKA); tyrosine kinase FYN; and Tau tubulin kinase TTBK1 [72]. Tau phosphatases include protein phosphatase-1, -2A, and -5 (PP1, PP2A, PP5, respectively). In transgenic mouse models, reduced PP2A activity and a parallel increase in CDK5/GSK3 activity cause early onset of tau hyperphosphorylation and accumulation [73,74].

In neurites, phosphorylation and ubiquitination of tau NFTs are found at comparable proportions and in proximal sites. While tau ubiquitination is a fundamental PTM that promotes tau targeting for degradation, ubiquitinated tau has also been shown to accumulate in both early and intermediate stages of disease. In the AD brain, mapping of aggregated tau ubiquitination identified 28 Lys residues, reported to be associated with tau conformational changes and increased phosphorylation [45,75]. A more intact conformation of the N-terminus of tau seems to facilitate ubiquitination, whereas late truncated and more compressed misfolding of the N-terminus may not be permissive to ubiquitination, suggesting that ubiquitination occurs at stages of disease when tau is either full length or only truncated at Asp421 (by caspase 3). Thus, the timing of tau ubiquitination, and changes in conformation and proteolytic processing are markers of the evolution of tau pathology in AD. Nevertheless, tau can adopt different conformations is different tauopathies, which will influence the pattern of ubiquitination, particularly for residues along the proline-rich region (Lys163, Lys180, Lys190, Lys224, Lys228, Lys234, Lys240) [75]. While current analysis methods can detect tau ubiquitination, they cannot definitively distinguish between mono- or poly-ubiquitination, or the type of polyubiquitin linkage (Lys6, Lys11, Lys27, Lys33, Lys48, and Lys63) associated with tau conformational changes and NFT formation [76]. Progress in this area of research will be relevant to also determine how the process of tau proteolysis by the proteasome or autophagy is affected by aberrant tau ubiquitination in disease.

Physiological tau acetylation is mediated by the p300/CREB-binding protein (CBP) HAT, and some studies have also reported that tau has intrinsic acetyltransferase activity [50,52,77,78]. Acetylation of tau inhibits binding to microtubules, increasing microtubule dynamics [79], and occurs in sites that can potentially compete with ubiquitination and affect tau degradation propensity (Figure 1c). In disease, tau hyper-acetylation is considered a major contributing factor to tau pathogenicity [80], and is found predominantly increased on the microtubule-binding regions (Lys174, Lys274, Lys280, Lys281) by p300, which is also elevated in disease [50,77]. Acetylation of tau can increase the propensity for oligomerization and aggregation, particularly in P-Tau, and can promote neuronal toxicity by interfering with microtubule assembly and neuronal plasticity, obstructing AMPA receptor trafficking and synapse potentiation [50,81,82]. Acetylated tau has served as a marker for pathology in vivo and as a diagnostic biomarker for AD [52]. Importantly, some studies have also reported that hyper- vs. hypo-acetylation of particular tau Lys residues can modulate tau clearance and NFT assembly [51,78]. For instance, acetylation of Lys259/353IGS motifs was found to be protective by inhibiting the phosphorylation of nearby Ser residues that otherwise would promote aggregation [78]. This is an important observation that highlights the importance of balance and competition between PTMs in rendering tau pathogenic. Tau Lys274 acetylation is one of the most described in disease, but its effect at the molecular level is not yet clear.

Proteolysis is another protein modification that can substantially increase tau aggregation propensity, toxicity, and transcellular propagation, and therefore disease progression. Tau truncation together with abnormal phosphorylation can induce conformational changes and misfolding of the N-terminus [83,84,85], and the truncated fragments can seed aggregation or spread between neurons. Tau proteolytic fragments are found in the CSF and plasma of tauopathy patients, and are now considered novel disease progression biomarkers [86]. As such, modulation of tau proteolysis is a potential therapeutic strategy aimed at selectively blocking tau proteases that generate disease-associated fragments. On the other hand, tau proteolysis can play a beneficial role from the perspective of increased processing and protein clearance. A detailed overview of known tau proteases and role in disease can be found elsewhere [86]. Briefly, the two most studied tau cleavage sites are at the C-terminus: one at Asp421 cleaved by caspase-3, and the other at Glu391 cleaved by an unknown enzyme [87,88,89]. Caspase-3 cleavage at Asp421-Ser422 results in the production of the NTF Tau-421 (or Tau-C) fragment found in many tauopathies, which is highly aggregation prone. The caspase-6-generated fragment at Asp402-Thr403 results in the production of the NTF Tau-402, which has been used as a CSF biomarker for AD. Increased caspase-6 is found in aging, and in sporadic and familial forms of AD. Calpain-1 and calpain-2 have also been reported to cleave tau at Lys44-Glu45 and Arg230-Thr231, respectively, but with opposite consequences for the fragments produced on synaptic function disruption and neuronal toxicity. In mice, the calpain-2-generated Arg230-Thr231 tau fragment impaired anterograde and retrograde organelle transport, induced synaptic loss, and caused profound hippocampal pyramidal cell death. Cathepsin D has also been shown to cleave tau at several residues and is a lysosomal proteolysis enzyme that contributes to tau clearance via autophagy. Finally, the puromycin-sensitive aminopeptidase (PSA) is a predominantly cytoplasmic neuronal enzyme enriched in the brain, identified as a genetic modifier of tau-associated toxicity in model systems, and found to be increased in FTD patient brains, so far suggesting a neuro-protective role in tauopathy and other proteinopathies [90].

### 2.2. Tau Misfolding and Phase Transition

Recent breakthroughs in the resolution of disease-associated tau conformations and fibrils have made evident that distinct conformers of monomeric misfolded tau can assemble into aggregates and hydrogen-bonded protofilaments that are packed in different ways to form fibrils that originate unique molecular signatures for each tauopathy, frequently referred to as disease-specific strains [91,92]. Growing evidence shows that brains from different individuals with the same tauopathy reveal the same tau strains, and therefore each disease is characterized by its own unique tau fold [91,93,94]. Tau isoform composition, PTMs, and interactions with cofactors determine which structures are formed in the brain, which are different from in vitro tau aggregates. Notably, insoluble tau isolated from FTD brains is rarely composed of a single conformational entity and is typically a mixture of up to three different conformers that together may give rise to distinct neurological phenotypes. This suggests that diversity of tau folds is intrinsic to the pathogenesis of each form of tauopathy and that effective therapeutic interventions will need to address evolving repertoires of misfolded tau species rather than singular static molecular targets. Moreover, understanding filament formation and deciphering the atomic coordinates of tau filaments will be useful for the design of aggregation inhibitors, as well as diagnostic PET (positron emission tomography) ligands specific for each tauopathy.

In most cases, tau aggregation encompasses the transition from a disordered unfolded monomeric state into a highly ordered fibrillar conformation (Figure 1b) [95,96]. This process first involves tau monomers’ transition into aggregation-competent intermediate conformations that then irreversibly assemble into compacted higher order aggregates. Due to a high energy barrier for the formation of the intermediate conformation, tau aggregation is a slow process, driven by conditions that favor compact conformations of tau [97]. In recent years, it has been recognized that pathogenic tau undergoes liquid–liquid phase separation (LLPS), which describes the protein’s progressive condensation into discrete assemblies that dynamically exchange biomolecules with its surroundings [98,99]. Phase-separated tau is thought to be the main template for aggregation [100,101,102,103], as these tau droplets transition from a gel-like into an aggregate state of lower energy and grow into non-spherical solid aggregates. There might also be a physiological role for tau LLPS that may include regulation of stress granule formation and nucleation of microtubule polymerization [104,105,106]. There are still many unanswered questions on what drives tau LLPS and what non-tau molecular factors are involved. Autosomal dominant tau mutations can significantly enhance tau phase separation and oligomerization but result in droplets that display slower dynamics compared to wild-type tau. This indicates that prolonged phase separation leads to more static tau structures and facilitates progressive accumulation of “stable” tau oligomeric forms that are directly linked to mechanisms of tau toxicity [100]. That is, the mutation makes tau more susceptible to pathogenic conformational changes by extended phase separation and time-dependent adoption of toxic conformations and formation of stable oligomers. Tau-P301L, one of the most disease-prevalent mutations, showed the greatest propensity for oligomer formation, following an extended phase separation and formation of non-filamentous toxic tau conformations. These findings suggest that tau phase separation represents a critical process by which tau pathogenic toxic conformations are adopted.

Electrostatic forces are predominant drivers for LLPS of intrinsically disordered proteins like tau [107], which is further facilitated by crowding agents, RNA [102], and PTMs, such as phosphorylation [35,108,109], that drive tau loss-of-function, promote aggregation, and present with distinct profiles across tauopathies [48,83,88,108,109]. Spanning along the molecule, phosphorylation of several Ser/Thr residues (Figure 1c) introduces negative charges, changes the electrostatic interactions along the polypeptide backbone, and enhances the kinetics of tau LLPS. The degree of hyper-phosphorylation determines the growth rate and extent of droplet formation [100]. Tau acetylation is also thought to have an effect on LLPS. The main sites of tau acetylation are concentrated in the positively charged central region of tau and can significantly alter tau electrostatic properties (Figure 1c). However, a study by Ferreon et al. [110] showed a dramatic reduction in acetylated tau droplet formation, suggesting that hyper-acetylation of tau in fact disfavors LLPS. This study puts forward a hypothesis where hyper-acetylation disfavors full-length tau LLPS by neutralizing the Lys positive charges, disrupting opposite-charge attractions that help support tau interactions and phase transition [110]. Even though acetylation per se may not drive aggregation, some of the tau Lys residues that are acetylated (Lys254, Lys311, Lys353) are also sites of ubiquitination [111]. Thus, acetylation will compete and prevent ubiquitination, and promote tau evasion from the ubiquitin-proteasome system and degradation, consequently promoting further tau accumulation. Furthermore, hypo-acetylated tau at key motifs in the microtubule-binding domains shows increased hyperphosphorylation, which drives tau aggregation [78]. So, indirectly, tau acetylation is still associated with increased tau aggregation. In support of this, AD mice treated with the deacetylase sirtuin 1 (SIRT1) showed a reduction in neuronal loss, whereas deletion of *SIRT1* enhanced tau pathogenicity [112].

The molecular forces driving LLPS and conformational changes among tau PTMs and mutations are not fully understood, but it is likely that both modifications change tau to facilitate stronger intermolecular interactions that underlie the enhanced liquid droplet formation in disease. Understanding the physical forces and biochemical changes driving tau oligomerization is critical to be able to design effective tau binders, either small-molecule aggregation inhibitors or antibodies with therapeutic potential.

### 2.3. Tau Spreading and the Prion-Like Model

In AD, tau pathology propagates in a temporal and sequential mode from medial temporal regions to the basal and lateral temporal cortices, inferior parietal cortices, posterior cingulate cortices, and other associated cortices [113]. Neurofibrillary tau appears to spread along neurons that are anatomically and synaptically connected, and these networks show the highest correlation with cognitive decline and clinical severity seen in patients [113,114]. An important discovery was that tau aggregates purified from patients’ brain tissue can template tau aggregation and accelerate propagation of pathology in transgenic cells and animals [115,116]. Because these amyloid assemblies can seed and template aggregation (“replication phenomenon”) of a homologous protein in naïve cells, an emerging view is that of tau propagation through the brain by a prion-like mechanism, where the conformation of the tau filaments determines its seeding characteristics [113,117,118,119,120,121]. In favor of a ‘spreading’ model of pathology, tau is also detected in the brain interstitial fluid (ISF) of transgenic mice [122] and in the CSF of AD patients, where both tau and P-Tau are elevated, suggesting a role for extracellular tau in disease pathogenesis and/or a consequence of disease progression. While a fraction of tau found in the CSF may be the result of passive tau release from dying cells [123], multiple lines of evidence suggest that active cellular processes are also involved in tau secretion.

Multiple vesicle-mediated and non-vesicular processes have been linked to the transmission of tau pathological species [54,118,124]. Extracellular tau can be found both as a free protein or in vesicles, such as exosomes and ectosomes. Recent studies have shown that tau can be released to the extracellular fluid both in vivo and in cultured cells, and that this is stimulated by neuronal activity [125,126]. Secretion of tau at the synapses during normal neuronal activity may be a physiological process where monomeric tau could have some yet undetermined signaling role and be unrelated to tau propagation. In animal transgenic disease models, a great fraction of pathological tau appears to be localized at synapses, and synaptosomes isolated from human AD brains were shown to contain more aggregated P-Tau than those of healthy controls [124]. Neuronal depolarization-associated presynaptic activity and calcium fluctuations are associated with the release of ectosomes and exosomes vesicles, which can facilitate trans-synaptic tau spreading into recipient post-synaptic cells [120,127,128]. This was demonstrated in cultured neurons as well as with human AD CSF, where tau-containing exosomes were released into the extracellular space and were then taken up by synaptically connected neurons [127]. Tau has also been found in ectosomes from cultured neurons, mouse brain ISF, and healthy controls’ CSF, suggesting that secretion in ectosomes could also be part of a physiological phenomenon, while exosomal tau secretion may prevail under pathological conditions [124]. Misfolded tau is also delivered as cargo to endo-lysosomes during cell stress and overload of the proteasome system, for secretion to the extracellular space through fusion with the plasma membrane (vesicle-free form). A recent study uncovered that several components of the endosomal sorting complexes required for transport (ESCRT) machinery, including charged multivesicular body protein CHMP2A and CHMP2B, are involved in tau propagation [129]. These findings suggest that “leakiness” of the endo-lysosomal compartment can contribute to propagation of aggregating tau.

Extracellular tau can be taken up by neurons through several mechanisms, including receptor mediated endocytosis (e.g., heparan sulfate proteoglycans (HSPGs) receptor-mediated uptake), phagocytosis, and pinocytosis [130,131]. Work led by Diamond et al. in mouse neural progenitor cells, utilizing tau-repeat domain fragment pre-formed fibrils, showed that tau uptake, in a similar way to prion proteins, occurs via binding to HSPGs on the cell surface [132]. Tau binding to HSPGs stimulates tau cellular uptake via pinocytosis (i.e., fluid phase endocytosis) into large intracellular vacuoles, and in turn this can be blocked by pinocytosis inhibitors or genetic knockdown of a key HSPG synthetic enzyme [130]. In this context, exostosin-2 (EXT2), a heparan sulfate-synthesizing enzyme, was found to play a role in the regulation of tau uptake in human cell lines [133,134]. To determine whether specific HSPG proteins or motifs mediate cellular entry of tau, Kampmann, Kosik et al. executed a CRISPR interference screen to identify modifiers of inter-cellular tau propagation [133]. Key regulators of this process were identified to be enzymes in the HSPG biosynthetic pathway. In particular, 6-O-sulfation of heparins was critical for tau–heparan sulfate interaction and competition or removal of these motifs from the cell surface reduced tau internalization [133]. The effect of 6-O-sulfation of heparins was validated in human CNS cell lines, human iPSC (induced pluripotent stem cell)-derived neurons, and mouse brain slices. This discovery can now develop into new strategies to halt tau transmission and possibly disease progression.

Low-density lipoprotein receptors (LDLRs) are known to work in conjunction with HSPGs [135], and so Rauch et al. [136] set out to investigate whether members of the LDLR family could modulate tau internalization. LRP1 (low-density lipoprotein receptor-related protein 1), a receptor highly expressed in neurons at the post-synaptic density [137], was identified to control tau uptake by cells via endocytosis. CRISPR silencing of *LRP1* in H4 neuroglioma cells almost completely blocked the uptake of full-length soluble monomeric tau (2N4R isoform), inhibited the uptake of tau oligomers, and reduced but did not completely inhibit the uptake of sonicated tau fibrils. The uptake of mutant tau and P-Tau was also affected by *LRP1* knockdown [136]. Based on cryogenic electron microscopy (cryo-EM) structures of tau fibrils [91,93], interaction of tau with LRP1 is expected to occur via two motifs: one within the microtubule-binding region, and one in the N-terminus or in the C-terminus [136]. In agreement with the working hypothesis that tau is propagated trans-synaptically, *LRP1* knockdown in human iPSC-derived neurons or in a tauopathy mouse model efficiently reduced the amount of internalized tau and significantly reduced tau spreading [136,138].

Pathological tau involved in intercellular transmission has multiple biochemical forms, including soluble monomeric, soluble oligomeric, insoluble aggregated, and fibrillar (Figure 1b) [120]. Many studies suggest that small tau oligomeric species are the most critical for pathological transmission and toxicity [139]. In a variety of cellular models, including human iPSC-derived neurons, tau uptake is a time-dependent process where monomeric, oligomeric, and sonicated fibrils are efficiently internalized but intact fibrils are not [133]. In particular, iPSC-derived neurons show a preference for smaller tau species whereas tau NFTs show almost no uptake. These observations suggest that the size of tau species is a critical factor for cellular uptake. A recent study by Kayed et al. [140] demonstrated that primary cortical neurons also show different efficiencies of tau species uptake. Soluble tau oligomers from AD brain were internalized via HSPG-mediated endocytosis, whereas internalization of PSP brain oligomers seemed to rely on HSPG-mediated as well as other pathways (e.g., clathrin mediated) [140]. Internalized exogenous tau oligomers were found to disrupt the autophagy–lysosomal pathway and to enhance the levels of P-Tau, which both were attenuated by *EXT2* knockdown. These findings again implicated HSPG-mediated endocytosis and EXT2 in tau uptake and suggest that these could be relevant pharmacological targets to prevent tau intercellular spreading [140].

A study by Dujardin et al. [69] showed that a key difference across AD cases of diverse clinical severity was a striking variability in tau spreading and seeding activity, hyperphosphorylation extent, and oligomerization state in postmortem tissue. In particular, tau seeding propensity showed a high correlation with the levels of oligomeric P-Tau in each brain. That is, a more rapid course of disease was associated with large soluble tau oligomers of P-tau [69]. With an elegant mass spectrometry analysis, the authors showed that specific phosphorylated tau residues were associated with the rate of disease progression and tau seeding activity, namely pThr231/Ser235 and pSer262. Interestingly, epitopes found in the CSF of preclinical stages of AD (pThr181, pThr217) did not show association with seeding activity. The results of this study suggest that P-Tau Thr181 and Thr217, found in CSF during early disease stages, are not necessarily the most pathogenic tau species, also underscoring the importance of soluble tau oligomeric assemblies over tau fibrils in AD. In light of the cryo-EM studies showing two predominant types of tau fibrils in the AD postmortem brain [91,141], it seems that before the formation of end-stage fibrils, there is high heterogeneity in tau oligomers and seeding capacity that show better correlation with disease severity.

In several diseases, tau also accumulates in astrocytes and oligodendrocytes (Table 1). The morphologies of glial tau aggregates vary from astrocytic plaques in CBD to tufted astrocytes in PSP, and oligodendroglial coiled bodies in both [15]. Although studies in the human brain have shown glial cell death as an early feature of neurodegeneration [142,143], the mechanism underlying the formation of glial tau pathology is less understood. Brain CBD-Tau and PSP-Tau were found to be propagation competent in neurons and glia (astrocytes, oligodendrocytes) in the brain of non-transgenic mice [144], suggesting that the formation of glial aggregates could be dependent on neuronal tau. Narasimhan et al. [145] showed that oligodendrocytes can develop tau pathology independent of neurons, but astrocytes cannot. In mice injected with human CBD-Tau or PSP-Tau, oligodendroglial tau aggregates propagated across the brain along white matter independently of neuronal axons and in the absence of neuronal tau pathology, whereas astrocytic tau aggregates did not. This suggests that glial tau pathology has significant functional consequences independent of neuronal pathology, with oligodendrocytes using their own processes for tau transmission [145]. Microglia may also play an important role. Postmortem studies revealed deposits of tau in reactive microglia, but because microglia do not express tau, this indicates that microglia had engulfed tau aggregates from the extracellular space or from dead neuronal debris. Once internalized, microglia have the ability to degrade tau under a certain threshold, above which excess tau leads to microglial dysfunction. While a correlation between microglia activation and spreading of tau pathology has been reported, the mechanism by which microglia transfer pathological tau to neurons is unknown [124].

The extent to which observations from in vitro systems, neuronal cell models, and transgenic animals accurately represent disease progression in the human brain and fulfill the criteria for prion classification has been vigorously debated. Moreover, extracellular tau levels in CSF and blood increase in AD, but these changes appear to be better associated with Aβ deposition than with tau pathology, because increased extracellular tau has not been observed in other tauopathies [146]. Most extracellular tau species, in both control and AD CSF, are truncated before the microtubule domains and are not seed competent. Thus, as there is limited evidence in humans for extracellular tau capable of seeding aggregation, it is only a hypothesis that prion-like spread underlies the progressive accumulation of tau pathology across neuropathologies. It is also not absolutely clear whether the spread of tau pathology is due to neuronal connectivity or due to the differential vulnerability of specific neurons or regions to tau pathology, or both. A growing understanding of the cellular processes involved in the transmission of seed-competent tau, as well as the identity of these species, will be critical for the development of effective therapeutic agents, chemical or immunological (Figure 2), that bind to tau, blocking aggregation nucleation and propagation [147,148,149]. Likewise, identification of the mechanism and receptors involved in transcellular propagation may lead to the development of specific small-molecule inhibitors of tau secretion and uptake by neurons.

### 2.4. Protein Degradation Failure

The cellular quality control system is responsible for maintaining a healthy proteome and relies on coordinated action of a multitude of chaperone functions and the proteolytic systems [150]. The autophagy-lysosomal pathway and the ubiquitin proteasome system (UPS) mediate the degradation of these abnormal proteins (reviewed in [111,151]). In tauopathies, these systems’ efficiency is challenged and overloaded by the time-dependent accumulation of misfolded aggregation-prone tau, which occurs in parallel with a reported age-dependent decline in proteostasis maintenance [152,153,154,155,156,157,158]. Other proteases that also process tau, including calpain, caspases 3, 6, and 9, can lead to limited tau proteolysis, which instead contributes to more toxicity through the generation of amyloidogenic fragments [159].

The multi-subunit ATP-dependent protease 26S proteasome catalyzes the selective degradation of proteins signaled for degradation by ubiquitination [160,161]. Ubiquitination of tau was first described upon the discovery that NFTs contained ubiquitin [162,163], and that tau in AD PHFs was ubiquitinated at Lys254, Lys257, Lys311, and Lys317 through Lys48-linked ubiquitination [164]. With further advances in mass spectrometry technologies, tau poly-ubiquitination at Lys6, Lys11, and Lys63 was also demonstrated [45,165]. Later, tau was shown to be a substrate of the E3 ubiquitin ligases CHIP and TRAF6 [166,167,168]. While Lys48-linked polyubiquitin chains are regarded to be a signal for proteasomal degradation, the majority of AD PHF-Tau was reported to be mono-ubiquitinated, which constitutes a weaker signal for degradation [164,169,170]. Consequently, the majority of PHF-Tau is not efficiently targeted for proteasomal degradation, suggesting that it can instead contribute to tau aggregation [171,172]. Work by David et al. [173] showed that recombinant tau could be degraded by the 20S proteasome core particle in vitro in a conformation-dependent manner, and that this process originated the formation of stable tau intermediates. However, incubation of 26S proteasomes with mutant or aggregating tau from AD brain caused inhibition of the barrel protease and “clogged” the entrance of other substrates into the proteasome [174,175]. Moreover, tau disaggregase chaperones have recently been shown to further generate species that are seeding competent, instead of facilitating proteasome degradation [176]. In AD triple transgenic mice, clearance of soluble tau by the UPS was observed only in the early stages of disease but not upon accumulation of hyperphosphorylated tau aggregates [177]. Not surprisingly, tau aggregation is a major factor in impeding its clearance, because the narrow pore of the proteasome barrel precludes the entrance of oligomeric and aggregated proteins [178]. Several studies have now shown that PHFs and NFTs can associate with proteasomes, impair its activity, and lead to overall higher levels of undegraded ubiquitinated proteins [175,179]. Tau impairment of proteasome activity might be caused by a change in the quaternary structure of the 26S complex by tau aggregates, leading to profound deterioration of its organization, or by the attachment of a tau “fibrous blanket” to the proteasome, leading to attenuated activity. Other tau PTMs can also prevent degradation by the proteasome by preventing ubiquitin conjugation: hyper-phosphorylation of the repeat domain region competes directly with ubiquitination (Figure 1c); C-terminal Asp421 truncation by caspase 3 enhances Tau autophagy; and acetylation at Lys163, Lys174, and Lys180 delays tau processing by the proteasome [52,180,181]. The seemingly inefficient process of tau ubiquitination and proteasome degradation, together with accumulated ubiquitinated tau in PHFs and NFTs from AD and FTD brain, has led to the argument that the proteasome has little influence on tau degradation [182,183,184,185]. Although evidence clearly shows that tau can be, in specific conditions, a substrate of the proteasome, and that in vivo administration of small molecules that enhance proteasome activity (e.g., PKA activators) promotes clearance of abnormal tau and improves cognition [186], tau PHFs can interfere directly with proteasomal activity, further amplifying tau toxicity [175,187]. In-depth characterization of tau ubiquitination and the multitude of enzymes involved in substrate recognition, modification, and processing for entrance in the proteasome barrel is crucial for the effort of developing enhancers of the UPS proteolytic function in healthy aging and in disease.

In turn, degradation of aggregated proteins by the autophagy-lysosomal pathway has been extensively reported [188,189]. The essential components of the autophagy proteolytic system are the lysosomes, single-membrane vesicles that contain in their acidic pH lumen a large variety of hydrolases [190,191]. The low lysosomal pH has been proposed to facilitate unfolding of substrate proteins, followed by the action of endo- and exoproteases that process proteins into smaller peptides and free amino acids. A review on the different types of autophagy and cargo specificity can be found elsewhere [151,191,192]. Autophagy not only plays a role in removing aggregated proteins that are too large to be degraded by the UPS [193,194,195], but it also appears to be a primary route of clearance for endogenous tau in healthy neurons [182,184,185,196]. Tau is a long-lived protein, and as such is predicted to be degraded in lysosomes. In the AD brain, tau immunoreactivity co-localizes with lysosomes, and inhibition of lysosomal proteases in the rat brain induces the formation of tau tangles [153,197]. Additionally, unlike the proteasome, autophagy degrades tau regardless of PTMs. The presence of ubiquitin on the surface of tau inclusions has been shown to actually facilitate the recruitment of components of the autophagic machinery to the aggregates, leading to the formation of engulphing autophagosome. Aggregates can also be recruited by cargo recognition proteins, such as p62, which can interact directly with ubiquitin moieties and with LC3, one of the essential autophagy proteins that associate with the autophagosome membrane [198]. Interference with the clearance of phagolysosomes, rather than induction of autophagy itself, gives rise to patterns of pathology resembling AD [153]. Tauopathy disruption of autophagy can also encompass deficiencies in autophagosome formation, cargo recognition, autophagosome mobilization toward lysosomes, autophagosome–lysosome fusion, and inefficient degradation of the autophagic cargo once delivered to lysosomes [151,199]. For this reason, even though tau aggregates can be positive for ubiquitin and interact with cargo recognition molecules, they still fail to be “efficiently incorporated” into the autophagic system [200]. A study by Wang et al. showed that chaperone-mediated autophagy is also involved in the proteolytic processing of tau, at least in cell models [185]. Tau contains two targeting motif homologs (336QVEVK340 and 347KDRVQ351) in the microtubule-binding domains, which are involved in the direct delivery of cytosolic tau by chaperones to lysosomal membranes. However, these motifs are also aggregation cores, ubiquitination sites, and oxidation sites, which can interfere with this process. In this study, induction of chaperone-mediated autophagy in mutant tau cells led to the generation of more aggregation-prone fragments as mutant tau failed to fully translocate into the lysosomes, perhaps due to aberrant PTMs. Several additional studies have shown evidence of abnormal autophagy-lysosomal function in the brain of tauopathy patients, as well as in animal and cellular models, where accumulation of autophagic vesicles, lysosomes, and tau correlate with neuronal toxicity [152,153,155,157,201,202] and enhanced tau aggregation [203,204]. Notably, lysosomes have also been implicated in the mechanism of exocytosis of selected tau species, mediated by the transcription factor EB (TFEB), a master regulator of lysosomal biogenesis [122]. TFEB loss of function in the Tau-P301S mouse model reduced the levels of ISF tau, and in cultured primary neurons, it reduced the amount of tau in the extracellular media, while endogenous tau was not affected. This work showed that TFEB-mediated exocytosis of non-seeding-competent tau could serve as a clearance mechanism to reduce the intracellular tau burden under pathological conditions [122]. Supporting this concept, prevention of TFEB-mediated tau exocytosis enhanced neuronal pathology. This study importantly showed that effective tau therapies should avoid targeting certain molecular processes and calls for the need to better understand the role of tau exocytosis in mechanisms of tau toxicity. Whether autophagy-lysosomal impairment is a contributor or a consequence of tau pathogenicity is still unclear [154,159]. In other words, deficient autophagy can be causal to disease, or its failure can be secondary to alterations in other quality control mechanisms. Nonetheless, this pathway seems to be amenable to genetic and pharmacological enhancement, which can reduce tau levels and aggregation, mitigating spreading and neuronal loss [193,201,202,205,206,207,208,209,210,211,212,213]. Based on these observations, enhancement of autophagy-lysosomal function can have critically relevant therapeutic implications [159,195,202,212,213,214,215,216,217].

### 2.5. Neuronal Stress Responses to Tau and Stress Granules Formation

How neurons respond to the accumulation of misfolded tau monomers, oligomers, and fibrils remains poorly understood, but it is undoubtably linked to the toxicity events that follow. Molecular chaperones are responsible for maintaining protein homeostasis, i.e., folding, function, and solubility, but age-associated decline in the function of the chaperone network has been increasingly held responsible for the incidence of aging diseases of protein aggregation. Since individuals carrying tau mutations only develop symptoms relatively late in life, it is possible that neuronal chaperones are capable of preventing tau toxicity for years. Interactions between tau and a number of chaperones have been documented, indicating that multiple players are involved, namely HSPB1, HSP27, HSP90, and HSP70/HSC70 interact with tau in ways that are likely relevant to disease [218]. For instance, HSPB1 is a small heat shock protein that acts as a holdase to prevent protein aggregation; HSC70 is a constitutively expressed member of the HSP70 family of chaperones that promotes refolding of misfolded proteins. Interestingly, these two chaperones are predicted to affect tau aggregation through different but complementary mechanisms. While HSPB1 is considered a “first responder” to delay formation of toxic oligomers [219], HSC70 interacts with oligomers and later species in the fibril formation pathway to protect from toxic conformations and promote degradation [218]. Meanwhile, the chaperone HSP90 has been proposed to promote tau proteasomal degradation through unclear mechanisms [220]. Importantly, the interaction of aggregating tau with chaperones may also have deleterious effects on proteostasis by sequestering chaperones and interfering with their other cellular functions [221]. Pharmacological enhancement of the cytosolic heat shock response, leading to upregulation of chaperones that potentially can keep tau folding vs. oligomerization under surveillance, is a therapeutic strategy under study by several groups [222].

Neurons have additional mechanisms of coping with protein stress, and one such mechanism involves a shift in protein synthesis. In this context, stress granule (SG) formation is mediated by RNA-binding proteins that regulate RNA translation, trafficking, sequestration, and degradation. In turn, these RNA-binding proteins are strongly regulated by signaling cascades integrating RNA translation and protein synthesis [223]. SGs are membrane-less organelles thought to be created through a phase separation-like process. Nucleation by core RNA-binding protein, such as TIA1, is followed by recruitment of secondary RNA-binding proteins to form a mature SG, which is a key component for stress-induced translational suppression. SGs normally accumulate in the soma and dendrites as small insoluble complexes and promote cell survival by blocking translation of non-essential mRNAs, allocating resources to protein (re)folding and sequestration of pro-apoptotic proteins [223]. In moderation, this stress response is likely protective for neurons. However, overactive SG formation and dysfunction of neuronal RNA-binding proteins can have deleterious consequences similar to the accumulation of protein aggregates [224,225]. Therefore, SGs are now considered pathology markers in many neurodegenerative disorders [226]. How does tau relate to formation of SG? Under basal conditions, tau is present in dendrites only at low levels, whereas in disease, misfolded, oligomeric and aggregated tau redistributes from the axon to the somatodendritic compartment [227,228]. Here, tau seems to associate with RNA-binding proteins and facilitate the formation of larger than normal SGs that serve as pathological seeds, where misfolded tau finds other low-complexity proteins with aggregating domains, forming a nucleating core for further aggregation [225,229]. Tau-mediated SG formation is associated with a shift in protein synthesis and an increase in sequestration of RNA-binding proteins in the cytoplasm [230]. The SG marker and nucleating protein TIA1 contains prion-like poly-glycine-rich domains, which further promote aggregation. TIA-1 has been found in AD and FTD hyperphosphorylated NFTs, in increasing amounts with increasing disease severity [231,232]. Increased Tau–TIA1 interaction is observed in cells expressing hyper-phosphorylated tau, and this has led to the hypothesis that TIA-1 plays a role in tau recruitment to SGs and in doing so modulates tau toxicity. In agreement, *TIA1* knockdown cells show significantly less tau-positive and tau-TIA-1-positive SGs [233]. Thus, TIA1 seems to be a key player in tau-SG formation, as even a partial knockdown strongly reduced tau recruitment to SGs. Evidence suggests that the biology of tau and TIA1 are linked in disease, with both proteins accumulating in the brain over disease progression [232].

The positive link between tau oligomerization, cell toxicity, and SG formation has been demonstrated in cell models and in vivo [230,234,235]. SGs may also play a role in cell-to-cell propagation of Tau species, possibly serving as a pathological seed. A study by Brunello and Huttunen in HEK293 cells showed that contrary to overexpressed cytosolic tau, internalized extracellular hyperphosphorylated tau (pinocytosis) was associated with cytosolic SGs and reduced viability of the recipient cells [233]. Regarding how internalized tau affects SG dynamics, this study showed that whereas SGs normally disassemble and resolve within a few hours from the disappearance of the stressful stimulus that promoted their formation [223], in cells exposed to tau-conditioned media, SGs remain for an abnormally extended period of time, with a significant amount of SGs positive for both tau and TIA1. Although there was not a significant increase in cell death, the cells were more sensitive to secondary stressors (e.g., subtoxic dose 30 nM of rotenone), implying that tau recruitment to SGs increased vulnerability to subsequent stressful events [233]. Importantly, modifications in tau chemistry or structure must occur during the secretion-uptake process, because only internalized tau species were able to effectively interact with TIA1 and promote formation and stabilization of SGs. In a parallel study, Vanderweyde et al. showed that interaction of tau with TIA1 in the brain affects SG formation and influences tau aggregation [230]. Tau interaction with TIA1 further enhances tau misfolding and assembly at the site of SGs, resulting in the stimulation of apoptotic markers in primary neurons. In this system, *TIA1* knockdown prevented tau misfolding and toxicity, and highlighted a role for RNA-binding proteins as therapeutic targets in tauopathies [230]. Treatment of primary neurons with compounds that prevent SG formation, such as translational inhibitors (e.g., cycloheximide) and kinase inhibitors (GSK3β and p38 inhibitors), reduced tau-SG formation [230].

Proteomic studies in mouse cortical brain tissue revealed that the loss of tau abrogated interaction of TIA1 with multiple RNA-binding proteins linked to RNA metabolism, suggesting that tau protein is required for normal physiological interaction of TIA1 with the RNA metabolism machinery and SG formation [230]. These results suggest that part of this tau pathological mechanism derives from an intimate physiological interaction with RNA-binding proteins: Tau must play a role in neuronal RNA-binding protein biology, regulating RNA transport and translation during stress. This raises the possibility that pathophysiology in tauopathies is strongly associated with dysfunction of RNA-binding proteins, and tau relocation to the somatodendritic compartment facilitates interaction of TIA1 with other SG proteins, and facilitates SG formation and translational stress response activation [227,228,230,236].

### 2.6. Disruption of Mitochondrial Function and ER Unfolded Protein Stress Response

Neuronal function is energy consuming, and mitochondria are the main energy source providing ATP through oxidative phosphorylation. Mitochondria also regulate many cell survival and death mechanisms and safeguard neuronal survival from a variety of stresses during the long neuronal lifespan. Not surprisingly, mitochondria are involved in the aging-associated decline in proteostasis, and disturbances in mitochondrial function are closely associated with mechanisms of neurodegeneration [237]. In the AD brain, there are extensive mitochondrial abnormalities and oxidative damage, consistent with changes in energy metabolism and impaired mitochondrial function that precede clinical onset and persist throughout the course of the disease [238,239]. Early analysis of AD brain biopsies showed ultrastructural mitochondrial damage in the vulnerable pyramidal neurons, with changes in the size, number, fission, fusion, and altered distribution along the neurons with a lower abundance in the processes [240,241,242]. These observations revealed that fragmented mitochondria are a pathological feature of AD, and possibly other tauopathies, contributing to energetic deficits and the generation of reactive oxygen species. Moreover, an uneven mitochondria distribution in the neuronal processes, due to the disruption of anterograde and/or retrograde transport, leaves large axonal and dendritic segments devoid of healthy mitochondria, which can impair the integrity and function of the neuron. Deficits in mitochondrial axonal transport have been attributed to the accumulation of hyper-phosphorylated tau and its somatodendritic relocalization in disease, which is expected to disrupt axonal transport through dynein and kinesin motor proteins, and compromise axonal anterograde transport [243,244,245,246,247]. In agreement with this, P-Tau reduction by GSK3β inhibition prevented deficits in the anterograde axonal transport of mitochondria in AD primary neurons [248]. Conversely, an increase in oxidative stress by mitochondrial dysfunction further promotes tau phosphorylation through upregulation of kinases, such as GSK3β [249]. In fact, mitochondrial and metabolic stress influence many signaling cascades regulating tau phosphorylation in disease [250]. Of note, mitochondria are also key targets of autophagy (mitophagy) in the brain, and growing evidence shows that autophagy-lysosomal dysfunction in tauopathy may further contribute to disruption of mitochondria recycling and homeostasis [156,251]. This leads to further accumulation of damaged mitochondria as evidenced by swollen appearances in electron microscopy images, in human AD biopsies, and in transgenic animal models of AD [252]. An accentuated decline in glucose consumption has been consistently found in the hippocampus and cortex of AD brains by PET imaging [237], and glucose hypometabolism became a standard biomarker of AD detection. The rate of oxygen metabolism was also found consistently decreased in the AD cortex, revealing a strong correlation with dysfunction of the mitochondrial electron transport chain and the severity of disease [237]. Several gene expression and histology studies in postmortem AD tissue have identified disruption of mitochondrial metabolic pathways, in particular downregulated expression of electron transport chain subunits (complex I and IV), TCA cycle, oxidative phosphorylation, and mitochondrial import genes, all representative of impaired energy metabolism in AD [253,254,255]. However, it still remains to be resolved how chronic and relatively mild impairment of mitochondrial function (e.g., 15–50% decrease in complex I or IV activity reported) can cause the changes observed in the AD and other tauopathies [238]. In summary, a “healthy pool” of mitochondria supports neuronal activity by providing needed energy and by protecting against neuronal activity and age-dependent oxidative damage. In turn, disruption of mitochondrial function, impaired bioenergetics, and increased oxidative stress are considered neurodegeneration contributors, as well as amplifiers of neuronal dysfunction. In this regard, understanding the multitude of mitochondrial mechanisms altered in tauopathies is critical.

The endoplasmic reticulum (ER) plays a crucial role in the folding and transport of cellular proteins that enter the secretory pathway, such as membrane proteins involved in synaptic function. In disease, prolonged ER stress has been implicated in increased neuronal vulnerability to cell death [256,257]. ER transmembrane-lumen stress sensors and transcription factors are critical responders to neuronal stress by the accumulation of misfolded proteins and altered physiological demands, mounting the ER unfolded protein response (UPR) [258]. The UPR relies on three major sensors of protein misfolding [257,259,260], each represented by a transmembrane protein: the inositol-requiring enzyme IRE1, the protein kinase RNA-like ER kinase PERK, and the activating transcription factor ATF6. Upon ER stress, each of these UPR arms is activated through a specific mechanism. PERK is highly expressed in the brain [261], and when activated, it phosphorylates the eukaryotic translation initiation factor 2 (eIF2α), which mediates a global reduction in protein synthesis. In AD, UPR is activated and leads to upregulation of P-eIF2α and of the ER HSP70 chaperone (BiP), which are found co-localized with P-Tau. In fact, neurons with an active UPR show an exponential increment in NFTs and a positive correlation with neuropathology [262]. Importantly, in the neurodegeneration context, there is, in addition to tau, an increase in the load of misfolded proteins, cellular stressors, and redox changes that also contribute to UPR activation. In fact, a direct tau causal role has not been established. Moreover, different tau mutations, and animal and cellular models of tauopathy have historically shown variable degrees of neuronal UPR activation as determined by markers’, such as P-PERK, levels associated with P-Tau accumulation [262]. Despite some discrepancies, the available data provide sufficient evidence to support the hypothesis that the progressive accumulation of tau in the neuronal cytosol is capable of inducing ER stress. A plausible scenario includes long-term events of tau misfolding, accumulation, and aggregation, where the “buffering” capacity of the cytosolic chaperones HSP70-HSP90 is overwhelmed, protein degradation is insufficient/inefficient, and tau accumulation in the cytosol depletes energy production and disrupts redox homeostasis, ultimately inducing ER stress. Several studies have also established a link between ER stress, GSK3β activation, and increased tau phosphorylation [256,263,264], and autophagy-lysosomal dysregulation, which altogether amplify toxicity leading to neuronal death [265]. Much research has focused on the genetic and pharmacological modulation of stress response pathways (cytosolic, ER, mitochondrial) with the goal of improving cellular and protein homeostasis maintenance in the context of chronic insults, such as misfolded and aggregating proteins, in neurodegenerative diseases [222,266].

### 2.7. Disruption of the RNA Splicing Machinery

In disease, somatodendritic tau co-localizes with TIA1, an RNA-binding protein that regulates SG formation and stimulates aggregation of additional RNA-binding proteins in the cytoplasm, including tau [223,230,232]. The fact that *TIA* knockdown in tau-P301S transgenic mice delays neurodegeneration, prolongs lifespan, and reduces the amount of SGs suggests that RNA-binding proteins directly affect tau pathogenicity [234]. The disease-modifying effect of *TIA1* reduction on tau has raised the hypothesis that dysfunction of RNA metabolism is an important contributor to pathology. That is, an increase in SG formation in tauopathies, and increased nucleation of RNA-binding proteins, disrupts their function as well as RNA splicing. In this context, disruption of RNA splicing has been observed in tauopathy mouse models and in the AD brain. As a consequence of alternative splicing errors, tauopathy neurons show functional defects associated with isoform-specific proteins. In a study led by Apicco et al. [267], the tauopathy mouse model expressing human Tau-P301S (PS19 mice) presented with downregulation of key RNA-splicing pathway components and altered mRNAs possibly associated with disease mechanisms. Disruption of RNA splicing in this model significantly affected the expression of synaptic and neurotransmitters expression genes, with half of these transcripts encoding proteins with known roles in glutamatergic synaptic function and calcium signaling [267]. For instance, flip GluR2-containing AMPA receptors were upregulated, which leads to slower receptor desensitization, rendering the cell more susceptible to excitotoxicity [268,269]. This study also showed that reduced expression of *TIA1* in the Tau-P301S mice rescued some of the disease-related changes in RNA splicing [267]. Further dissection of the genes mediating the protective effect of *TIA1* silencing identified factors known to regulate tau pathophysiology, such as FYN kinase, HSP90, and the E3 ubiquitin ligase CHIP. Other high-impact genes not previously associated with tau toxicity were also identified in the *TIA1* knockdown, including the proto-oncogene/transcription factor MYC and EGF receptor (EGFR) [267]. Pharmacological inhibition of either MYC or EGFR protected against toxicity in tau-overexpressing cells. Overall, the patterns of dysfunctional RNA splicing in AD brain and tauopathy models are largely overlapping and many are predicted to be regulated by TIA1. These findings further support the potential for therapeutic approaches targeting RNA-binding proteins, regulation of splicing factors, and SG formation across tauopathies.

### 2.8. The Role of Neural Inflammation in Neurodegeneration

The immune response is important for maintaining fundamental neuronal functions, such as long-term potentiation, neural plasticity, and neurogenesis [270]. However, immune and inflammatory responses can also accelerate the process of neurodegeneration [271,272]. Multiple transcriptomic and proteomic studies of the human brain from AD or other tauopathies (e.g., PSP) have identified clear dysregulation of immune response pathways [273,274,275,276]. Innate immunity and neuro-inflammatory changes in tauopathy are mainly characterized by release of inflammatory mediators, and changes in microglia and astroglia morphology, reactivity, distribution, and gene expression. All these factors contribute to disease progression and key roles have been attributed to TREM2, CD33, and CR1 [277,278,279,280]. Reactive microglia can induce astrogliosis through the production of specific cytokines (interleukin 1 alpha or IL1α and tumor necrosis factor alpha or TNFα) that promote cell death, with the hippocampus showing particular vulnerability to the increase of the proinflammatory cytokine interleukin-1β (IL-1β) [281,282]. Studies in AD mice indicate that microglial neurodegenerative phenotypes include downregulation of homeostatic genes (*P2ry12, Tmem119, Cx3cr1*), and a parallel upregulation of genes, such as *Apoe*, *Tyrobp*, and *Trem2* [278,283]. In AD, microglial-expressed TREM2 (triggering receptor expressed on myeloid cells 2) signaling seems to be protective against the spreading of tau pathology [284]. TREM2 signaling is associated with downstream regulation of cell proliferation and survival, suppression of inflammatory cytokine production, and facilitation of metabolic ATP production [285]. In line with this observation, rare variants in the TREM2 gene increase disease risk by 2–4-fold, possibly by impairing the response of microglia. In AD CSF, soluble TREM2 levels have been shown to correlate fairly well with total and P-Tau Thr181 levels. AD patients that also harbor the TREM2-R47H variant display even higher levels of total tau and P-Tau Thr181 in CSF compared to non-carriers [286,287,288], which is indicative of exacerbated pathogenic tau burden in the brain and neuronal loss [289].

Primary tauopathies also manifest prominent neuroinflammatory gene expression signatures. RNA-sequencing studies in transgenic mice have highlighted early upregulation of inflammatory processes and downregulation of synaptic function genes preceding behavioral phenotypes, suggesting that tau has a direct impact on microglial activation and synaptic dysfunction [290,291]. Little is known about the role of TREM2 in the specific context of primary tauopathies, where accumulation of pathological tau species is the main driver of cell death. Sayed et al. [292] found that, in the Tau-P301S (PS19) transgenic mouse model, TREM2 knockout was protective against tau-mediated microglial reactivity and atrophy but that TREM2 haplo-insufficiency or the AD-associated TREM2-R47H variant led to elevated expression of proinflammatory markers, exacerbated atrophy, and increased tau pathology [279]. However, when Bemiller et al. [293] crossed the TREM2 knockout mouse with a less aggressive mouse model of tauopathy, the authors observed a decrease in microgliosis in TREM2-deficient mice, as observed in TREM2-deficient PS19 mice, but reported that the complete deletion of TREM2-exacerbated tau pathology. More recently, Gratuze et al. [284] showed that in TREM2-deficient Tau-P301S mice (PS19-T2R47H), glial fibrillary acidic protein (GFAP) gene expression, a marker of reactive astrocytes, was significantly decreased. These observations suggested that the TREM2-R47H variant (mainly associated with AD) strongly reduced microglial activation and astrogliosis in the setting of tauopathy. Consequently, there was a reduction in P-Tau staining, attenuated brain atrophy and synapse loss, and reduced microglia reactivity. These findings support that impaired TREM2 signaling reduces microglia-mediated neurodegeneration in the setting of primary tauopathy. Moreover, reduced TREM2 signaling reduces microglial conversion to a proinflammatory phagocytic state and is protective against neurodegeneration in the setting of advanced tauopathy [284]. Overall, these studies have led to the hypothesis that TREM2 and its downstream signaling have distinct effects depending on disease stage and aggressiveness. During early stages of tau accumulation in the absence of neurodegeneration, a decrease in TREM2 function exacerbates tau pathology, while the complete loss of TREM2 function in advanced disease stages (PS19 mice) protects from neurodegeneration.

Cumulative evidence shows that neuroinflammation is a common hallmark of AD and other neurodegenerative diseases and is proposed to actively contribute to pathogenesis. Most observations show a high correlation between pathological tau, neuronal loss, and immune response, emphasizing the need to understand how tau directly or indirectly promotes the release of inflammatory mediators. Modulation of risk factors and targeting of these immune mechanisms could lead to future therapeutic strategies for AD and other tauopathies.

## 3. Tau-Directed Therapeutics

The rate of failure in drug development for tauopathies is relatively high. Promising preclinical data has supported the development of multiple experimental therapies for tauopathy, but showing a pharmacodynamic effect in transgenic animal models has had poor translatability into clinical efficacy. For AD, 99% of all drug trials have failed partly due to limited bioavailability, poor blood–brain barrier (BBB) penetration, low cell permeability, and reduced drug half-life. The BBB still constitutes, and for good reason, a formidable obstacle to ensuring that any drug reaches its target in the CNS and therefore, an effective therapy will have to have a super affinity for tau and/or be administered at high doses, which becomes exceptionally difficult and expensive. Alternatives include intrathecal approaches, which can circumvent the BBB but have an immensely high cost of treatment and other medical implications associated that will render the medicinal application worldwide almost prohibitive. Using small molecules that cross the BBB is an attractive and possibly optimal approach, but so far, the majority of experimental tau drugs with positive preclinical data have clear limitations of BBB permeability. Another significant challenge for tau therapeutics is the demonstration of selective target engagement and unvalidated biomarkers for pharmacodynamics and a modifier effect on disease progression of primary tauopathies. Although robust assays can measure some forms of tau found in CSF and blood, these tau species do not necessarily represent the relevant pathological tau found in the brain parenchyma. Several PET ligands are being developed and tested but not without challenges. All these aspects highlight the need for profound changes in drug development approaches. An effective therapeutic agent is the result of a multi-variable effort that combines the right target, drug, biomarker, participant, and clinical trial (as reviewed in [294]). The right target represents the biologic process most relevant for tauopathy, with proven function in disease pathophysiology and greater representation in disease than in normal physiological function and is a non-redundant process necessary for neuronal survival. The main proposed targets and pathways relevant for tauopathies were described in the section above. So far, apart from the use of combination therapy in AD to target the cholinergic system that has some effect on cognitive function [295], no other target has been shown to have patient benefit. A challenge in developing small molecules that target tau and lead to effective disease-modifying therapeutics is the insufficient understanding of disease mechanisms, as well as the lack of a well-defined tau fold for active molecular binding in disease [214,296,297]. Moreover, a drug for tauopathy must have appropriate pharmacokinetic (PK) and pharmacodynamic (PD) parameters, ability to penetrate the BBB, efficacy demonstrated in animals, and, importantly, in patient-derived ex vivo neurons, and acceptable toxicity. Achieving appropriate BBB penetration is a major hurdle for many experimental therapeutics [298]. The human BBB has p-glycoprotein transporters and other multidrug resistance channels (MDRs) that may not be present in the animal models employed preclinically and that for this reason do not adequately predict human CNS entry. Despite the still critical limitations from an incomplete understanding of disease mechanisms, a lack of reliable clinical biomarkers, variable drug effects on less-then optimal mouse models, and limited implementation of humanized cellular models of tauopathy, the field has seen groundbreaking advances in multiple fronts for development of tau therapeutics [299,300]. At the time of this review, 24 therapeutics towards tau have been tested in clinical trials Phase 1 or later, with 15 agents currently in active development based on publicly available data (see [301] for a review). Here, we will highlight some of the main current approaches showing the most progress in experimental tau-directed therapeutics (Figure 2).

### 3.1. Modulators of MAPT Expression

Gene editing techniques are increasingly used in drug discovery research, [302], particularly when the genetic cause(s) of disease is known. However, with the recognition that late-life sporadic neurodegenerative diseases frequently have more than one contributing pathology, identifying a single molecular therapeutic target whose manipulation is efficacious in all affected individuals may not be straightforward [303]. Additionally, and as highlighted in previous sections, given the diverse roles of tau in the human brain, complete *MAPT* knockdown has been approached cautiously. Although in many preclinical mouse models, complete *MAPT* knockout has no overt phenotypes and may even be protective against seizures, in older animals, behavioral and cognitive changes have suggested that tau is necessary for normal brain function [304]. Additionally, low tau levels have been described in association with dementia lacking distinctive histopathology, such as sporadic FTD [305], pointing to some discrepancies regarding the silencing of *MAPT* expression in the human brain and possible detrimental long-term consequences. Nonetheless, promising new strategies focusing on anti-sense oligonucleotides (ASOs) to decrease tau expression have led to multiple reports of reversal of tau pathology in mouse and non-human primate models (Figure 2) [147,306,307]. ASO-based clinical treatments have demonstrated dramatic success in other neurodegenerative diseases and in transgenic preclinical mice models, suggesting that this approach would be viable in tauopathy patients [147]. In transgenic mice expressing Tau-P301S (PS19 model), ASO treatment caused a 50% reduction in *MPAT* mRNA levels and reversed tau aggregation, with a concomitant decrease in the rate of hippocampal atrophy, neuronal loss, and behavioral deficits. These encouraging findings led to an ongoing Phase 1/2 trial of anti-tau ASOs (IONIS-MAPTRx) in mild AD (Clinicaltrials.gov ID NCT03186989). Additionally, as of recently, a new safety, tolerability, and pharmacokinetic study of multiple ascending doses of another MAPT ASO (NIO752) has been launched by Novartis for PSP trials (Clinicaltrials.gov ID NCT04539041). Given this initial success, the same group of investigators are leading the preclinical development of 4R-Tau specific ASOs [308].

### 3.2. Inhibitors of Tau PTMs

Tau PTMs have a strong effect on the regulation of tau function and propensity for aggregation and, therefore, were considered early on as relevant therapeutic targets (Figure 1c and Figure 2). Regarding aberrant P-Tau, there are a number of kinases that mediate tau phosphorylation and that have been proposed as putative drug targets [214,309]. These are Ser/Thr protein kinases and Tyr protein kinases, and examples include GSK3, CDK5, MAPK, PKA, CaMKII, and TTBK1 [310]. However, even if phosphorylation plays a role in tau toxicity, developing tau-specific and safe kinase inhibitors is extremely challenging particularly for long-term treatment in tauopathies. Moreover, it is still unclear which kinase is responsible for each P-Tau residue (Figure 1c), and whether its function is specific or redundant. Kinases also have many roles and numerous substrates, so even in the event that an inhibitor is specific, the unintended off-target (non-tau) effects may still represent an important liability. Borrowing from the oncology field, initial drug screens for kinase inhibitors were straightforward, and several kinase inhibitors with an effect on tau phosphorylation were identified. The first kinase inhibitors tested targeted GSK3β, which is also a regulator of cellular differentiation, growth, motility, and apoptosis. Lithium, an FDA-approved mood stabilizer used to treat bipolar disorder, was found to reduce tau hyperphosphorylation and aggregation in Tau-P301L transgenic mice via a mechanism that was dependent on GSK3β inhibition [311]. This led to Phase 2 trials in AD, but no effect was found on cognition, mood, or CSF P-Tau biomarkers of the participants. In a later trial in patients with PSP or CBS, lithium was poorly tolerated. Valproate (Depakote) is another small-molecule FDA-approved mood stabilizer and anti-epileptic, also found to inhibit GSK3β and rescue behavioral phenotypes in AD mice [312]. Unfortunately, in a Phase 3 AD trial, treatment with valproate resulted in accelerated brain atrophy and cognitive impairment, with significant toxicity. A later study in PSP patients showed no difference in disease progression, with possible worsening on measures of gait. Valproate is no longer in clinical development for treatment of tauopathies, and available evidence recommends against its use. It should be noted that both lithium and valproate are non-specific for GSK3β, and thus observed toxicity may be due to off-target effects. It is also a possibility that GSK3β was not inhibited to the extent needed to significantly reduce tau phosphorylation for clinical benefit. Later, tideglusib was introduced as a novel GSK3β inhibitor. In the double transgenic mouse model of AD (expressing human APP and tau), tideglusib reduced tau phosphorylation, decreased amyloid deposition, rescued neuronal loss, and improved cognition [313]. However, in a Phase 2 trial in mild-to-moderate AD patients, it did not demonstrate any reduction in the rate of cognitive or functional decline [314]. FYN, another kinase implicated in tau phosphorylation, is proposed to be overactive in AD, leading to tau hyperphosphorylation and synaptic loss [315]. A small-molecule FYN inhibitor called saracatinib (AZD0530) showed promising results in AD mice, reducing tau aggregation and rescuing synaptic function [316]. However, in a Phase 2 AD trial, saracatinib showed no effect on primary or secondary outcomes, and gastro-intestinal side effects led to discontinuation in a quart of participants. The most recent kinase inhibitor tested in clinical trials was Nilotinib, a selective BCR-ABL kinase inhibitor that is FDA-approved for chronic myeloid leukemia. ABL is a tyrosine kinase that phosphorylates tau on the residue Tyr394, leading to increased tau aggregation into PHFs [317]. A Nilotinib positive effect on tauopathy phenotypes may also be explained by an indirect effect on autophagy activation and tau clearance [318]. In a small Phase 1/2 trial in patients with Parkinsonism dementia, nilotinib reduced CSF levels of tau and amyloid [319]. Based on these results, a Phase 2 trial in AD patients is underway, with a primary outcome of safety and tolerability (Clinicaltrials.gov ID NCT02947893). Noticeably, identification of kinase inhibitors with a verified mechanism of action specificity is very challenging, and so far, preclinical mouse models have not been predictive enough and do not provide enough information on extent of reduction of tau phosphorylation needed for a positive outcome in clinical trials. Additionally, animal models have not adequality represented the negative and off-target effects of these small molecules.

Activation of tau phosphatases, in particular protein phosphatase 2A (PP2A), has been proposed as an alternative strategy to kinase inhibition for reducing tau phosphorylation. Drugs increasing the activity of PP2A, probably through modulation of endogenous proteins that inhibit PP2A activity, have the therapeutic potential for treating tauopathies [320], but no clinical trials with PP2A activators have yet been initiated.

Tau acetylation (Figure 1c) has been reported to compete with ubiquitination, reduce degradation of P-Tau, and contribute to somatodendritic relocalization and toxicity [52], but it has also been reported to interfere with tau phosphorylation and diminish aggregation [78]. The residue Lys174 was identified as an important acetylation site critical for tau homeostasis. In Tau-P301S (PS19) transgenic mice, treatment after disease onset with salsalate, a small-molecule anti-inflammatory agent that precedes the FDA approval process, reduced tau acetylation at Lys174, decreased aggregation, prevented hippocampal atrophy, and rescued memory deficits in the mice [79]. These findings led to two Phase 1/2 trials for salsalate. The trial on PSP patients showed no efficacy after 6 months of treatment. The trial on AD patients is still ongoing (Clinicaltrials.gov ID NCT03277573). This study might indicate whether targeting tau acetylation has therapeutic potential in FTD; however, the brain penetration of salsalate is limited (<3%) and therefore more potent brain-penetrating inhibitors of tau acetylation might be needed to conclusively test this hypothesis. Furthermore, hyperactive p300/CBP in disease is implicated with increased tau acetylation, and is associated with aberrant blockage of autophagy-lysosomal function and increased tau secretion in neurons of transgenic mice. Very recently, Gan et al. [321] identified a new p300 inhibitor that significantly blocked tau secretion in vitro and tau spreading in vivo, with a reduction of tau accumulation and pathology.

On the other end of the spectrum, O-GlcNAc modification of tau (Figure 1c), which is the attachment of N-acetylglucosamine (GlcNAc) moieties to Ser/Thr residues, inhibits tau toxic self-assembly [322]. Tau O-GlcNAcylation can affect tau aggregation by blocking hyperphosphorylation by kinases. In the human AD brain, levels of O-GlcNAcylation were found to be reduced by 50% compared to healthy controls, and this inversely correlated with tau hyperphosphorylation, supporting O-GlcNAcase (OGA) inhibition in disease and that an increase in O-GlcNAcylation could have therapeutic relevance [323]. In support of this premise, the OGA inhibitor Thiamet-G was found to reduce the levels of pathologic tau aggregates in P301L transgenic mice [324]. Subsequently, the small-molecule OGA inhibitor MK-8719 showed similar effects in transgenic mouse models. A Phase 2 clinical trial for PSP was planned but was never initiated and this compound has since been discontinued [325]. Meanwhile, another OGA inhibitor, ASN120290 (ASN-561), was developed and showed positive outcomes in Tau-P301S transgenic mice, increasing the levels of O-GlcNAcylated tau by more than two-fold, with a parallel decrease in P-Tau. ASN120290 has been awarded orphan drug status and will soon commence clinical trials for PSP.

### 3.3. Tau Aggregation Inhibitors

Tau aggregation inhibitors are therapeutic agents that target the aggregating properties of the tau molecule and consequent gain-of-toxic function (Figure 2) [326,327,328]. Highly specific binders and inhibitors of tau aggregation that also block the seeding propensity of tau will undoubtedly contribute to the prevention of tau pathology in the brain. Small MW compounds have been developed to block tau–tau binding and inhibit the formation of oligomers and fibrillization. While some have reached clinical trials, several inhibitors have also shown toxic profiles in vivo early on (reviewed in [326,329]). Aggregation inhibitors fall into two mechanistic classes based on the type of interaction with tau. Covalent aggregation inhibitors are agents that either covalently modify tau directly or foster formation of covalent bonds within or between tau proteins to yield aggregation-incompetent conformations. These agents have a higher binding affinity for tau monomers but seem to be able to interact with all tau species. Examples include natural polyphenols (e.g., oleocanthal, oleuropein aglycone) and redox-active compounds (e.g., methylene blue). Although, covalent mechanisms of tau aggregation inhibition are predicted to have low utility in vivo, methylene blue is still the most advanced aggregation inhibitor in clinical trials. This compound was initially developed in the late 1800s for treatment of malaria and was later found to modulate oxidation of cysteines and disrupt tau–tau bonds, showing beneficial outcomes in animal models when given prior to symptoms onset. More recently, studies have shown that methylene blue can also reduce tau aggregation by promoting increased clearance of tau by autophagy and by the UPS [210]. In different transgenic tauopathy mouse models, methylene blue treatment reduced either insoluble P-Tau or mainly soluble tau, delayed cognitive decline, and improved behavioral phenotypes [329,330]. However, it did not reverse pre-existing NFT pathology. Contradictory results in vivo, with a lack of effect reducing P-Tau aggregation have diminished researchers’ enthusiasm for further development. A second-generation stabilized derivative of methylene blue, called LMTM, quickly entered the clinical trials route, but, despite high expectations [331], two different Phase 3 AD and FTD trials failed to produce beneficial outcomes or delay disease progression [332]. It was not tested whether LMTM had an effect on accumulation or aggregation of tau, or whether it changed the levels of pre-existing tau NFTs. Another covalent inhibitor, named N744, was found to inhibit in vitro fibrillization of full-length 4R tau and to promote disaggregation of pre-formed tau PHFs [333]. However, when at high concentrations, N744 can also form aggregates that enhance tau fibrillization, preventing its use in vivo [334]. Interestingly, for other similar dye-like compounds, it is the formation of aggregated compound that results in the inhibition of fibril formation [335]. A number of anthraquinones, including the anticancer drugs daunorubicin and adriamycin, have been identified as inhibitors of tau aggregation [336], which can also induce disassembly of pre-formed tau fibrils in neuronal cell models. In all cases, the molecules were more effective in disaggregating shorter tau fragments than full-length tau [337]. The Mandelkow team has subsequently identified a series of tau aggregation inhibitors, namely phenylthiazolyl-hydrazide (PTH) compounds [338,339]. It is unclear if covalent interaction with tau is a general property of this class of inhibitors. For many other aggregation inhibitors found to bind tau in vitro, there is still lack of evidence for efficacy in vivo. The second class of tau aggregation inhibitors are non-covalent agents that interact transiently with tau species, mainly with natively unfolded tau monomers [326,327,329]. Examples include molecules that render tau aggregation-incompetent (e.g., curcumin), molecules that block the formation of cross-β-sheet structures, and molecules that drive the formation of non-aggregating SDS-stable oligomers (e.g., phthalocyanine tetrasulfonate, Congo red derivatives, rhodanine). The Mandelkow team also identified a rhodanine series of tau aggregation inhibitors [338,339], and elaborated on the key structural elements within this series that were important for activity, which included disaggregation of pre-formed short tau filaments in neuronal cells.

Although an inhibitor of tau assembly has conceptual appeal, it remains to be demonstrated whether any of the existing candidate inhibitors consistently reduce tau aggregation in vivo. Moreover, it is also unclear whether reduction of somatodendritic tau inclusions will result in rescue of cognitive decline. Generation of such proof-of-principle data in vivo will require that compounds have appropriate chemical and biological properties, including good pharmacokinetic behavior, adequate CNS exposure, and low toxicity. Unfortunately, many of the existing tau aggregation inhibitors have chemical or biological characteristics that are likely to preclude them from being tested in preclinical tau models for a lack of good BBB penetration, unsuitable half-lives, and poor safety. Another critical aspect is the concentration of compound that will be required to inhibit aggregation in vivo when for in vitro assays, the concentrations of inhibitor are approximately equimolar to the amount of tau. Finally, it will be important to understand how aggregation inhibitors interrupt tau fibril assembly and what other species of tau are generated by the treatment, since it is still unclear whether mature tau fibrils or smaller tau oligomers are most toxic. Growing evidence supports a role for the latter [340], and therefore interrupting tau fibrilization by increasing the pool of intermediate multimers may in fact exacerbate tau pathology.

### 3.4. Modulators of Tau Clearance by Autophagy and the Proteasome

The autophagy-lysosomal pathway is recognized as the main mechanism for clearance of protein aggregates that cannot be proteolyzed by the proteasome. Whether autophagy and lysosomal impairment are contributors or a consequence of tauopathy is unclear [154,159], but many studies have shown evidence that pharmacological enhancement of autophagy activity in patient-derived neurons and in vivo can reduce oligomeric and aggregated tau, mitigate tau neuronal transmission, and reduce cell loss, supporting a role for autophagy modulators in therapeutics [159,193,206,207,208,211,212,213,215,216,217]. Several activators of autophagy have been put forward [159,212,214,215], but none have yet shown efficacy in the human brain at patient-tolerated doses, or successful outcome in clinical trials.

Autophagy stimulation by trehalose in the transgenic Tau-P301S mouse model significantly reduced tau inclusions in the brain, improving neuronal survival in the cerebral cortex and the brainstem [202]. Trehalose failed to activate autophagy in the spinal cord and had no effect on the motor impairment of Tau-P301S mice. These findings provided evidence in favor of tau degradation by pharmacological activation of autophagy in vivo. Studies in mouse models with rapamycin, an inhibitor of mTOR, showed a reduction in pathogenic tau levels and improved cognition by upregulation of autophagy [341]. Long-term inhibition of mTOR by rapamycin or latrepirdine also prevented AD-like cognitive deficits and lowered tau NFTs [205,209,342]. We recently described pharmacological upregulation of autophagy in tauopathy patient iPSC-derived neurons that rescued tau phenotypes [213]. The lead small molecules OSI-027, AZD2014, and AZD8055 are orally available, potent, and specific mTOR inhibitors, which in human neurons had a stronger effect on tau clearance than rapamycin, without affecting viability. Compound treatment downregulated P-Tau and insoluble tau, consequently reducing tau-mediated neuronal stress vulnerability. The most notable finding in this study was the discovery that a single dose 24-h treatment caused a reduction of tau for 12–16 days post-treatment without loss of cell viability or integrity, and across independent neuronal models. This is relevant because, in tumor cells, organ transplant studies, and clinical trials, mTOR inhibitors have shown a plethora of side-effects. Although this poses an obstacle for treatment of older patients with neurodegenerative diseases, adverse effects are usually dose and frequency dependent, and reversible upon treatment interruption [343]. This study therefore proposes that these adverse effects could be counter-balanced by an intermittent dosing regimen, on account of a prolonged drug effect. However, in tauopathies, autophagy induction does not necessarily have a straightforward beneficial outcome. Research has shown that in disease, induction of new autophagosome formation is not necessarily impaired, but that the lysosomal-dependent proteolytic system is the main cause of disruption, leading to “backed up” accumulation of autophagosomes carrying misfolded proteins [344]. These studies point to impaired lysosomal proteolytic function as the origin of autolysosome malfunction in AD pathogenesis [345]. Therefore, when considering autophagy modulation as a therapy, the lysosomal defect needs to be taken into account, because simply inducing autophagy without correcting clearance will not produce the desired outcome. In this regard, and luckily, in many cases, the pharmacological enhancement of autophagosome formation also activates the transcription factor EB (TFEB) that simultaneously coordinates lysosomal biogenesis as well as genes required for autophagosome formation, fulfilling the criteria of autophagy-lysosomal upregulation [346]. This also suggests that TFEB should be a therapeutic target. On the other hand, pharmacological treatments that improve the catalytic performance of lysosomal enzymes and reduce the load of autolysosomes should also rescue lysosomal function.

Modulation of the UPS in tauopathies is also appealing because chaperone proteins that regulate the UPS function also mediate tau folding. Several HSP90 inhibitors are already in clinical trials as anticancer agents, and when tested in cellular models promoted reduction of total tau and P-Tau [180,347]. However, many of these drugs cannot cross the BBB, and their specificity for misfolded pathogenic tau is uncertain. Several methods of proteasome activity enhancement have been thoroughly reviewed by others [186]. Briefly, in healthy neurons, increased neuronal activity is coupled to increased synaptic proteolysis by recruitment of proteasomes to dendritic spines and upregulation of proteolysis function via CaMKIIa phosphorylation of the proteasome ATPase subunit. In neurodegenerative diseases, reduced UPS activity and impaired synaptic proteolysis leads to accumulation of ubiquitinated P-Tau oligomers in the synaptoneurosomes. Phosphorylation of proteasome subunits (e.g., by PKA or CaMKIIa) has been reported to increase recruitment of 26S proteasomes to dendritic spines and enhance local protein degradation of aggregation-prone proteins in cell-based assays and in vivo models of FTD. Reversible proteasome phosphorylation leading to enhancement of its function has been documented in ex vivo human neurons, as well as in vivo in physiological and pathological animal models [186,348,349]. Another mechanism proposed to enhance proteasome substrate degradation is via inhibition of USP14, a proteasome-associated deubiquitinating enzyme. USP14 reduces proteolysis in a substrate-specific manner by rapidly trimming the tagging ubiquitin chains before the 26S proteasome can initiate degradation of that substrate [350]. As a result, deubiquitinated proteins are released from the 26S proteasome undigested. It is possible to enhance 26S proteasome-mediated proteolysis using small-molecule USP14 inhibitors, which have shown utility in cellular models [351,352]. Finally, phosphodiesterase enzymes (PDEs) have also been identified as regulators of the 26S proteasome, with possible therapeutic implications. Namely, a selective PDE4 inhibitor (rolipram) in a mouse AD model showed increased proteasome-mediated protein degradation [179,353]. Later, it was shown that PDE4 inhibition in fact stimulated PKA, and consequently phosphorylation of several subunits of 26S proteasome, rescuing proteasome function and degradation of tau in the Tau-P301L mouse model (rTg4510) [179]. Moreover, PDE10 inhibition and activation of PKA in the mouse striatum has also been shown to have beneficial effects by increasing proteasome function and reducing protein aggregates [186,353]. These studies suggest a new mechanism of action of PDEs, albeit in different models and disease contexts, whereby PDEs can modulate protein homeostasis through the regulation of proteasome function.

### 3.5. Anti-Tau Immunotherapy

Harnessing the immune system to clear protein aggregates is one of the earliest and most promising therapeutic strategies for AD, and numerous immunotherapy approaches targeting β-amyloid and tau have been developed [354]. Currently, there are nine different tau antibodies (passive immunotherapy) and two tau vaccines (active immunotherapy) in clinical trials and several more in late-stage preclinical development (Figure 3) [355]. Tau vaccines and humanized antibodies can target a variety of tau species either in the intracellular or extracellular space, recognize the N-terminus, C-terminus, the proline-rich region, or the microtubule-binding domains (Figure 1a), and can potentially selectively target only pathological tau conformations. Several mechanisms have been proposed to contribute to the efficacy of tau-targeted immunotherapies. Peripherally injected anti-tau antibodies have been shown to cross the BBB once they reach the brain and bind intracellular tau as well as extracellular seeds, therefore inhibiting the propagation of tau pathology. In addition to evidence for this mechanism that is still controversial, tau immunotherapies must first reach the brain. This process seems to be very limited according to the finding that antibody levels in the CNS only reach ~0.1% of peripheral levels [356]. Therefore, whether peripheral immunization will result in sufficient target engagement to change the course of tau pathology is still unknown. Even after reaching the brain, some anti-tau antibodies are not readily taken up into neurons, presumably because of their unfavorable charge and size, and are expected to exert an effect mainly in the extracellular space. Antibodies that do enter neurons do so via receptor-mediated uptake or endocytosis [355]. Once internalized, it is hypothesized that antibodies bind to misfolded tau and aggregates within the endosomal–lysosomal system, promote their disassembly, and enhance exposure to proteolysis in the lysosome. Alternatively, tau antibodies can potentially bind tau species in the cytosol, preventing release and spread, while increasing proteasomal degradation.

Active immunization works by injecting tau itself to induce a sustained autoimmune response, and if effective, it could potentially be used as a preventative agent in a manner similar to vaccines. Early studies using full-length human tau inoculation in mouse models showed high inflammation and enhancement of tau aggregation, limiting the enthusiasm for active immunization. Subsequent vaccination strategies have avoided full-length tau, employing protein fragments instead, which have not demonstrated an off-target immune response, though efficacy has yet to be proven. Active immunization trials include the ACI-35 vaccine developed by AC Immune and licensed to Janssen, consisting of 16 copies of a synthetic P-Tau fragment (Ser396/Ser404) anchored to a liposome—liposome-based vaccine (Figure 3) [357]. This vaccine aims to elicit an immune response to pathological conformers of P-Tau without also mounting autoimmune responses against physiological forms of tau. In transgenic Tau-P301L mice, vaccination led to the generation of polyclonal IgG antibodies specifically directed against P-Tau that bound to NFTs in the brain, improving some phenotypic parameters. Gliosis, T cell activation, and other inflammatory markers were reported as negative [358]. Similar results were obtained in non-human primates. ACI-35 is currently undergoing Phase 1 trials in mild-to-moderate AD patients and Phase 2 safety trials. Another active immunization trial with AADvac-1, developed by Axon Neuroscience (Figure 3), consists of a synthetic peptide derived from tau amino acids 294–305, which was conceived to trigger an immune response directed toward pathological N-terminus truncated tau species [359]. The AADvac-1 vaccine is undergoing Phase 1/2 trials in AD and progressive non-fluent aphasia (form of PPA).

Several tau-targeted passive immunotherapy approaches, which involve the administration of an anti-tau antibody, are also in development and in clinical trials (Figure 3) [355]. The AC Immune antibody Semorinemab targeting extracellular tau was recently tested at Phase 1 in healthy volunteers and AD patients. The goal for this antibody was to reduce microglial activation leading to detrimental inflammatory responses, and was reported to bind the N-terminus of all six isoforms of human tau, monomeric and oligomeric forms, regardless of PTMs [360]. Antibody treatment reduced brain pathology in Tau-P301L mice. Semorinemab is now undergoing Phase 2 for AD. Passive immunization with tau humanized IgG4 monoclonal antibody Gosuranemab is directed against extracellular N-terminus tau (Figure 3). This tau form was proposed to be involved in the secretion and spread of pathology, and in mouse FTD models, the antibody was reported to rescue tau toxicity [361]. In 2017, Biogen took Gosuranemab though a Phase 1 trial in healthy volunteers and AD patients. A Phase 2 trial in PSP patients has recently been discontinued (Clinicaltrials.gov ID NCT03068468). The C2N 8E12 humanized antibody (ABBV-8E12, Figure 3), developed by C2N Diagnostics and AbbVie, binds to the tau N-terminus and recognizes an aggregated extracellular form of pathological tau and does not rely on uptake into neurons for efficacy [118,362]. In cell-based assays, the antibody blocked tau seeding as well as the uptake of AD-derived tau [363]. In Tau-P301S transgenic mice, antibody treatment reduced the brain NFT load, microgliosis, brain atrophy, and behavior [149]. Although it has been discontinued at Phase 2 in PSP trials, it is still under testing at Phase 2 for AD [361].

In general, there is still not enough preclinical data showing that any form of tau immunotherapy is suitably optimized or that targeting of the tau species it was designed for is sufficient to prevent neurodegeneration [146]. Recent cumulative research results have made evident that several features of a tau antibody can have critical impact on its mechanism of action and overall efficacy. These features include the tau epitope, antibody isotype, charge, affinity, and size (reviewed in [355]), in addition to the decision of which tau conforms to the target, which undeniably brings back the question of which species of tau is most toxic, where toxicity occurs (intracellular vs. extracellularly), and in which tauopathy. In this context, a recent study by Goodwin et al. [146] highlights that immunotherapies that primarily engage extracellular tau may have limited efficacy. By comparing tau-targeting single-chain variable fragments (scFvs) and intrabodies (iBs) against pan-tau and P-Tau-specific epitopes, and respective efficacies in tau transgenic mice models, the study showed that (*i*) disease-modifying efficacy does not require antibody effector function, (*ii*) the intracellular targeting of tau with P-Tau-specific iBs is more effective than extracellular targeting with the scFvs, and (*iii*) the effect on tau pathology only resulted in modest disease modification as assessed by the delay of motor phenotypes [146]. Although this data suggests that intracellular targeting of tau might be more effective, it is unclear whether the tau epitopes targeted by this study’s antibodies corresponded to the ones present in these particular model’s extracellular tau seeds. However, recombinant adeno-associated viral vector (rAAV)-mediated delivery of select antibodies significantly reduced both pathology prior to the onset of neurodegeneration and delayed neurodegenerative phenotypes, demonstrating efficacy for the first time for a gene therapy immuno-approach for tau [146]. It is still questionable if the modest reduction in pathology observed would translate into sufficient disease modification in a human trial. The pathological progression of tauopathies is accompanied by the changes in tau conformation, and therefore epitopes, and as such certain antibodies may work better or worse based on the stage of disease and the localization of pathological tau. Anti-tau immunotherapies have shown great promise in animal and cellular models, and with the successful completion of many Phase 1 trials, safety concerns are currently low. However, antibody affinity and specificity, confounding aspects of organism immune response, and low effect in clinical pathological markers still posed considerable challenges [149,327,355,364,365,366].

### 3.6. Tau-Targeted Therapeutics with Bifunctional Degraders

Proteolysis-targeting chimeras (PROTACs) were first described by Crews and Deshaies in 2001 [367] as bifunctional molecules that hijack the ubiquitin proteasome system to close proximity of a target protein of interest in order to promote its ubiquitination and degradation by the proteasome. PROTAC-based protein degraders are an emerging strategy for ablating previously undruggable protein functions [368,369], by providing a mechanism to transform a non-functional protein binder into an effective degrader [369]. Structurally, these degraders are bivalent molecules where a binder of the protein of interest is linked via a short chain to an E3-ligase recruiting molecule, such as the cereblon (CRBN) binder immuno-modulatory drug thalidomide (Figure 4). This results in the formation of a ternary complex between the protein of interest, the degrader molecule, and the E3-ligase complex, and induces ubiquitination and subsequent proteasomal degradation of the protein (Figure 4) [370,371,372,373,374,375]. In contrast to classical drug pharmacology, no functional activity is necessary for degrading the protein of interest, and the mechanism of action is event driven, rather than occupancy driven, requiring lower drug concentrations. PROTACs show a catalytic behavior in their ability to induce proteasomal degradation at sub-stoichiometric levels [376]. The mechanism of cellular uptake for these molecules is not yet understood, but diverse molecular types of PROTACs have shown penetration in different cell types, which is suggestive of a passive process [377]. In 2018, Jiang et al. [378] reported a peptide PROTAC targeting tau and the Keap1-CUL3 E3 ubiquitin ligase. One of the peptidic PROTACs described was able to interact effectively with Keap1 and tau, showing cell penetration and inducing Tau degradation in different cell lines overexpressing tau. Although this was the first demonstration of the potential for PROTAC-mediated degradation of tau, the peptidic nature of these molecules would likely limit their therapeutic potential due to issues with instability and poor BBB penetration. Later, we too harnessed the PROTAC technology to transform one of the most clinically advanced tau PET tracers, flortaucipir (^18^F-T807), into the CRBN-recruiting (via pomalidomide) tau degrader QC-01-175 [379]. We demonstrated that a T807-derived degrader molecule preferentially degraded tau species in FTD patient-derived neuronal cell models expressing Tau-A152T or Tau-P301L, in a dose-dependent manner, thus rescuing tau-mediated neuronal stress vulnerability. Importantly, this tau degrader had a minimal effect on tau from wild-type control neurons and preferentially targeted tau species from FTD patient-derived neurons. Tau degradation was fully dependent on CRBN activity and tau binding, as well as proteasome function, whereas autophagy was not involved. These results propose that tau degraders may offer a promising, unprecedented opportunity for neutralizing the neurotoxic effects of tau in disease [379].

There are also challenges in this field, such as rational structure optimization, limited structural data, and the greatly anticipated in vivo and clinical data. Albeit promising, tau PROTAC-based degraders rely on the function of the UPS in a disease and aging context. However, the UPS has been shown to be associated with disease pathogenesis with a progressive reduction in function. This means that the use of tau degraders in neurodegenerative diseases like tauopathies may rely on combined therapy approaches that signal proteins for proteasome degradation and concomitantly enhance the proteasome proteolytic function. Unfortunately, interest in this pathway as a therapeutic target has lagged behind other clearance pathways, such as autophagy. Despite the remaining hurdles, PROTAC-based degraders are expected to soon become a new therapeutic category of drugs.

## 4. Concluding Remarks

Tau pathology is a hallmark of a number of genetic and aging diseases with growing incidence worldwide. Having a deeper understanding of disease mechanisms will ultimately contribute to finding a treatment that prevents the onset and relentless progression of neurodegeneration. The development of new biomarkers to complement current clinical measures of symptomology and brain function will be critical to enable measurement of target engagement in clinical trials, as it will help evaluate pharmacodynamic effects of treatment, and measure disease progression. Increasing knowledge regarding the structure of tau filaments in different forms of tauopathy (by cryogenic electron microscopy) provides valuable structural information that will contribute to an improved and rational design of specific tau aggregation inhibitors, tau ligands for PET imaging, and development of tau- and disease-specific degraders. A total of 60 clinical trials have been conducted to date with 24 different therapeutics targeting diverse proposed tauopathy mechanisms (Figure 3), an advance never before seen for tau and really “representing the dawn of a new age of drug development for Tau” [301]. However, there are still many unknowns regarding the multitude of tau-mediated mechanisms leading to neuronal loss of function and death, and there are also many barriers to be overcome until a tau therapeutic reaches the clinic. Therefore, the intellectual and financial investment in this field of research continues to increase. As a result, the rapid pace of progress in the research of tauopathy disease mechanisms and parallel development of novel experimental tau therapies targeting those mechanisms, especially over the past 5–10 years, is encouraging and suggests that we will eventually see a tau therapeutic in clinical use.

## Figures and Tables

**Figure 1 ijms-21-08948-f001:**
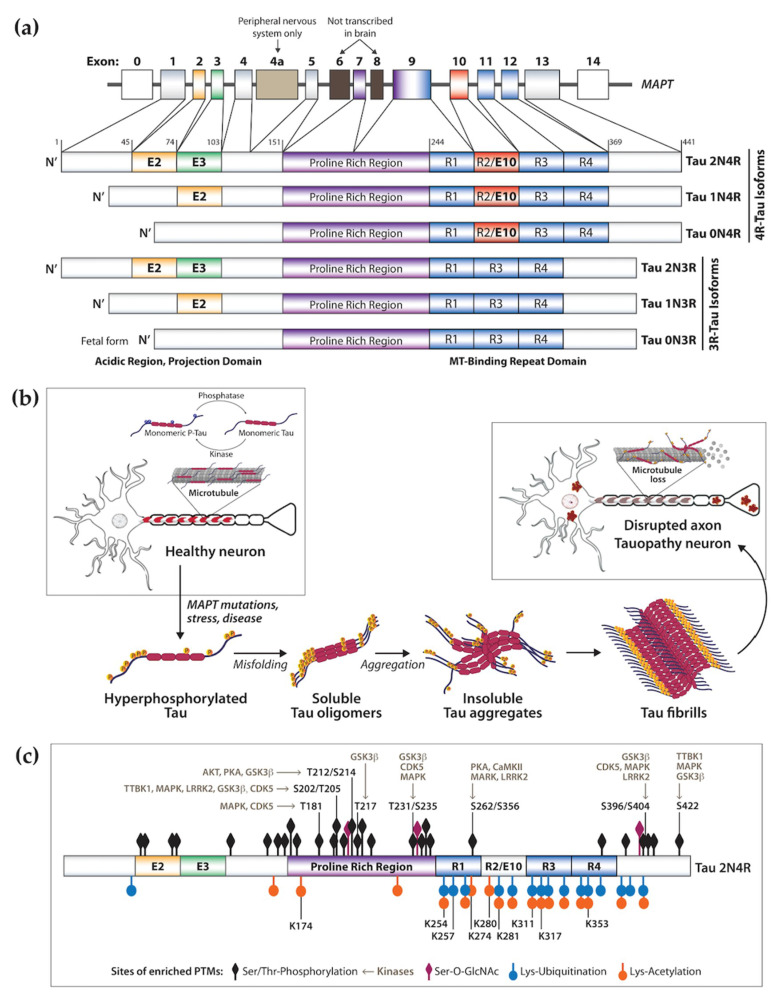
Human microtubule associated protein Tau physiological function and in disease. (**a**) Alternative splicing of the MAPT gene leads to developmentally regulated expression of six Tau isoforms, containing three (3R) or four (4R) microtubule (MT)-binding domains in the C-terminus, and zero, one or two N-terminus domains. (**b**) Simplified representation of Tau function as a regulator of microtubule stability and dynamics in human neurons. Tau binding is regulated by phosphorylation via the concerted action of kinases and phosphatases. In disease Tau becomes hyperphosphorylated and no longer binds microtubules, contributing to axonal dysfunction. Together with post-translational modification, Tau misfolding drives oligomerization and aggregation into larger order insoluble fibrils such as NFTs and PHFs found in the somatodendritic space and processes of CNS neurons. (**c**) Tau undergoes extensive post-translational modification (PTMs), which are exacerbated in disease. Indicated in the 2N4R Tau isoform are the locations of highest PTM density, including phosphorylation, acetylation, O-GlcNAcylation and ubiquitination. Also indicated are sites of phosphorylation prevalent in tauopathies and key regulatory kinases.

**Figure 2 ijms-21-08948-f002:**
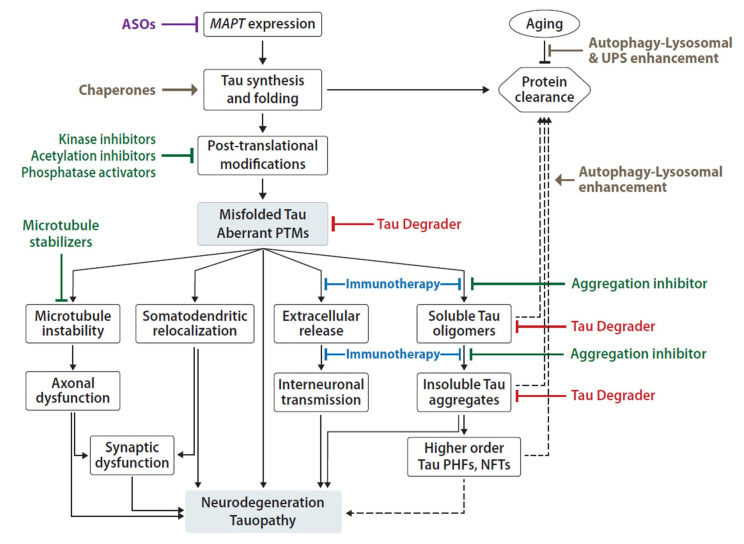
Summary of proposed mechanisms of Tau pathogenicity and corresponding experimental therapeutic approaches. Tau toxicity can be driven by loss-of-function leading to microtubule depolymerization and axonal transport disruption; and it can be driven by gain-of-function of aberrant Tau oligomers, aggregates and fibrils associated with neuronal toxicity, pathology spread and ultimately death. Current development of therapeutic agents include reduction of MAPT expression by ASOs (purple), small molecule (green) inhibitors of PTMs and aggregation, enhancement of Tau folding and/or clearance mechanisms (brown), Tau-specific degraders (red) and anti-Tau immunotherapies (blue). Solid arrows represent known and/or direct effects; dashed arrows represent indirect/proposed mechanisms; flat-ended connections represent inhibitory effect.

**Figure 3 ijms-21-08948-f003:**
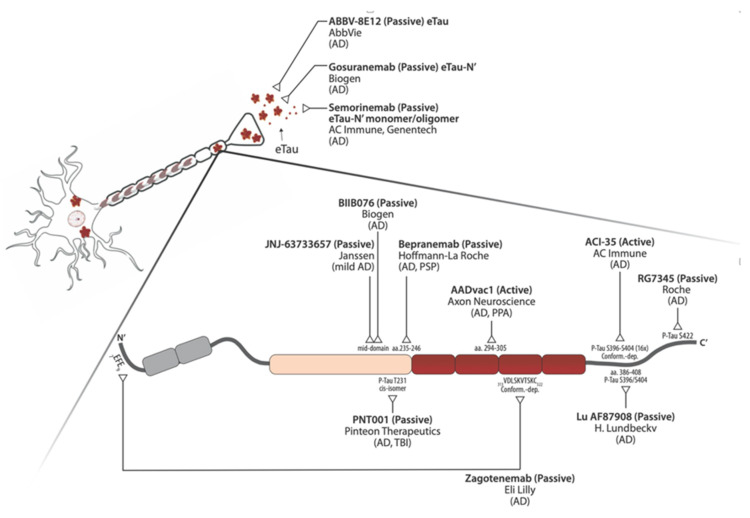
Overview of Tau immunotherapies in clinical trial. Indicated are passive and active Tau immunotherapy agents that have reached clinical trials Phase 1/2, for binding both intra and extracellular Tau in the CNS. The Roche RG7345 antibody has since been discontinued (Clinicaltrials.gov). Key: conform. -dep. refers to conformation-dependent antibody recognition of Tau, 16x means 16 Tau fragments repeats, and eTau refers to extracellular Tau.

**Figure 4 ijms-21-08948-f004:**
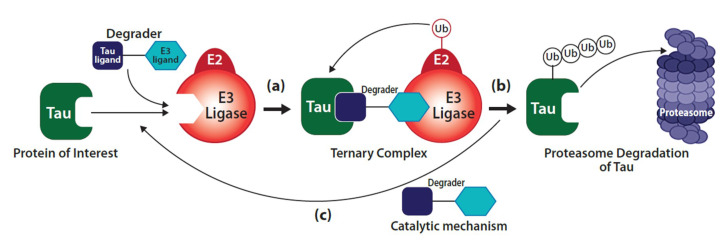
Proposed mechanism of action of Tau degraders, based on targeted protein degradation technology. (**a**) A degrader is a bifunctional molecule designed to preferentially recognize disease-associated Tau species and simultaneously engage an E3 ubiquitin ligase complex. (**b**) This leads to formation of a ternary complex that mediates Tau ubiquitination, targeting Tau for degradation by the proteasome. (**c**) It is proposed that this is a catalytic mechanism, and that upon Tau degradation, the degrader molecule is released and can associate with more Tau species, restarting the cycle.

**Table 1 ijms-21-08948-t001:** Summary and key features of primary and secondary tauopathies categorization.

	Clinical	Symptomology	Tau	Neuronal Pathology	Glia Pathology	Affected Brain Regions
**Primary Tauopathy**	Pick’s disease (PiD)	Behavioral change, social disinhibition, eating disorder, absent/late parkinsonism.	3R	Round cytoplasmic inclusions (Pick bodies), rare NFTs.	Ramified astrocytes.	Dentate gyrus of the hippocampus, frontal and temporal neocortical layers II, IV. Frontal, insular and anterior temporal cortices.
	Behavioral variant of FTD (bvFTD)	Behavioral disinhibition, apathy, empathy loss, compulsiveness, executive and cognitive dysfunction.	3R > 4R	Cytoplasmic NFTs, short dystrophic neurites.		Orbitofrontal, dorsolateral prefrontal, medial prefrontal cortices. Subcortical brain nuclei. Temporal-parietal lobes.
	Progressive supranuclear palsy (PSP)	Apathy, anxiety, sleep disturbance. Spectrum from pure motor to pure cognitive presentations.	4R	NFTs, neuropile threads.	Tufted astrocytes, somatodendritic coiled bodies.	Subthalamic nucleus, basal ganglia, brainstem. Posterior mesencephalic cortex.
	Corticobasal syndrome (CBS)	Asymmetric motor symptoms, apraxia, sensory impairment. Spectrum from pure motor to pure cognitive presentation.	4R	NFTs, neuropile threads, ballooned neurons, pleomorphic inclusions (pre-tangles).	Annular clusters of short fuzzy cell processes, astrocytic Tau plaques, argyrophilic inclusions.	Frontoparietal cortex, striatum, substantia nigra.
	Argyrophilic grain disease (AGD)	Personality change, emotional imbalance, memory failure.	4R	Argyrophilic grains, dendritic straight filaments and smooth tubules.	Thorn-shaped astrocytes, coiled bodies.	Medial temporal lobe, entorhinal cortex, hippocampus, amygdala.
	Aging-related Tau astrogliopathy (ARTAG)	Cognitive decline.	4R	-	Thorn-shaped and granular-fuzzy astrocytes.	Medial temporal lobe, lobar (frontal, parietal, occipital, lateral temporal), subcortical, brainstem.
	Globular glial tauopathy (GGT)	Behavior change, cognitive decline, motor neuron disease (Parkinsonism).	4R	-	Globular inclusions in astrocytes and oligodendrocytes.	White matter, limbic and isocortical regions. Hippocampus.
	Primary progressive aphasia (PPA)	Language deterioration, loss of semantic memory.	3R, 4R	NFTs, amyloid plaques	Globular astrocytic inclusions.	Anterior and temporal lobes, parietal lobe. Frontoinsular cortex
	Primary age-related tauopathy (PART)	Cognitive impairment.	3R, 4R	NFTs, neuropile threads	Medial temporal lobe.	Medial temporal lobe.
	Tangle-only dementia (TOD)	Late-onset dementia.	3R, 4R	Intracellular PHFs, NFTs and neuropil threads.		Hippocampus.
**Secondary Tauopathy**	Alzheimer’s disease (AD)	Memory loss, cognitive dysfunction, social behavior changes.	3R, 4R	NFTs, neuropile threads, neuritic plaques.		Entorhinal cortex, hippocampus, cerebral cortex.
	Chronic traumatic encephalop-athy (CTE)	Memory loss, confusion, personality/behavior changes. Motor decline.	3R, 4R	P-Tau aggregates around small vessels, TDP-43 cytoplasmic inclusions.	P-Tau aggregates around small vessels.	Cortical sulci, isocortex layers II–III, hippocampus, subcortical nuclei.

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
