# Peer review of "Tauopathies: Deciphering Disease Mechanisms to Develop Effective Therapies"

_ijms, 2020, doi:10.3390/ijms21238948_

Round 1

Reviewer 1 Report

  1. The authors have thoroughly reviewed the concept of tauopathy, its mechanisms, and treatment methods for disease etiology. Minor corrections are needed.
  2. It is important to include too general/obvious contents in very concise or may be excluded
  3. The number of references in the manuscript should be limited; only the most recent and most relevant references should be quoted.
  4. Abbreviations can only be used if they are used in several positions in the manuscript.
  5. In the abstract, line 4, there should be Alzheimer's Disease (AD)
  6. Concluding remarks should be compact

Author Response

We are grateful for the comments contributing to improvement of our manuscript, and for the opportunity (and time) to make the necessary revisions based upon the feedback received and requested changes.

  1. It is important to include too general/obvious contents in very concise or may be excluded.

We have extensively revised the manuscript to exclude general/obvious or repetitive content, as well as content that deviated from the main goal of this article, which is to focus on Tau mechanism of pathogenicity and, based on that knowledge, different approaches for Tau targeted therapeutics. As a result, we have reduced the body of text (excluding references) from 39 pages in the original version to 29 pages in the revised version.

  1. The number of references in the manuscript should be limited; only the most recent and most relevant references should be quoted.

We agree with the Reviewer, only the most recent references must be included, except for historical ground-breaking references that describe the original work. In the effort of respectfully select and acknowledge the most up-to-date work, we have now reduced the number of references from 587 in the original version to 436 in the revised manuscript.

  1. Abbreviations can only be used if they are used in several positions in the manuscript.

We thank the Reviewer for the feedback and now removed any abbreviations from the body of text and the ‘Abbreviations List’ that are not frequently used throughout the manuscript. These deletions are highlighted with the ‘Track Changes’ function of Microsoft Word.

  1. In the abstract, line 4, there should be Alzheimer's Disease (AD)

This has been modified as suggested.

  1. Concluding remarks should be compact

We agree with the Reviewer entirely. As part of the effort in reorganizing sections of the manuscript and eliminating unfocused paragraphs, we have now also significantly reduced the size of the ‘Concluding Remarks’, to now focus on a brief summary (single paragraph) of where the field stands and ongoing scientific developments that will contribute to near future progress.

Reviewer 2 Report

My major problem with the review is lack of focus. Authors are trying to give entire available information in the field rather focusing towards the review point.

My strong suggestion to the authors is try to make the review more meaningful towards the development of effective therapies with brief insight into understanding of disease mechanism in Tauopathies.

Authors need to work extensively to cutdown the unnecessary information and make it more focused with good interpretation in the field to move forward.

Author Response

We are grateful for the comments contributing to improvement of our manuscript, and for the opportunity to make the necessary revisions based upon the feedback received and requested changes.

We thank the Reviewer for the valuable input and appreciate the opportunity to extensively revise our manuscript based on this comment. We have reorganized some of the sections of the manuscript and excluded repetitive content, as well as content that deviated from the main goal of this article, focused on the mechanisms of Tau pathogenicity and respective approaches for Tau therapeutics. As a result of the effort in reorganizing sections of the manuscript and eliminating unfocused paragraphs and sections, we have reduced the body of text (excluding references) 39 pages in the original version to 29 pages in the revised version. We have also revised our references list to only include the most up-to-date and original work/historical references and have now reduced the number of references from 587 in the original version to 436 in the revised manuscript. We hope that with the changes made we have addressed the Reviewer’s major problems with the manuscript to make the body of text more attractive and focused to readers.

Round 2

Reviewer 2 Report

Even though review has its own importance in the field, still I feel the information in manuscript is not to the point. See below suggestions carefully and adjust accordingly.

"Deciphering Disease Mechanisms to Develop Effective Therapies"

I would like to see the direct message Related to the topic "disease mechanisms behind the Tauopathies that determine the therapeutic approaches".

These below sections are not needed to the review as these are taking away the focus point.

1.1. History, Socio-Economic Impact and Perspectives

1.2 Tau in Health and Disease (authors talked about all aspects including biomarker and imaging aspects in this section which is not required)

1.3. Subclasses of Tauopathies

(Authors can still present the figure and table by referring them in one or two sentences). Authors can introduce above three section in one or two sentences from below section

2. Molecular Mechanisms of Tau Pathology

Nine subsections in this must be crisp and should make sense to the following section  

3. Tau Directed Therapeutics

Author Response

Point 1. These below sections are not needed to the review as these are taking away the focus point.

1.1. History, Socio-Economic Impact and Perspectives; 1.2 Tau in Health and Disease (authors talked about all aspects including biomarker and imaging aspects in this section which is not required); 1.3. Subclasses of Tauopathies.

We thank the Reviewer for the thoughtful suggestions to make the content of the paper more straightforward. The Introduction section (1.) was extensively revised and shortened to make the content more “on point” as suggested. The following sub-sections were removed: 1.1. History, Socio-Economic Impact and Perspectives (originally on page 1); 1.2 Tau in Health and Disease (originally on page 2); 1.3. Subclasses of Tauopathies (originally on page 6).

The revised introductory section (pages 1 to 4, no sub-sections) is now a simpler introduction of Tau protein and necessary concepts to immediately be followed by mechanisms of disease (section 2.) and therapeutic approaches (section 3.). Table 1 and Figure 1 are still included.

Point 2. Molecular Mechanisms of Tau Pathology - Nine subsections in this must be crisp and should make sense to the following section.

The text in section 2 has been revised to make the content clear and linked to current efforts in drug discovery research (section 3). All non-essential content was removed, and every subsection makes a reference to the connection between the mechanism of disease described and possible therapeutic strategy.

Section 2.8 (Tau function in the nucleus and DNA damage) was removed because there is still not a clear causal link between Tau function in the nucleus and disease, nor focused therapeutics.
